# Tailoring high-energy storage NaNbO₃-based materials from antiferroelectric to relaxor states

Mao-Hua Zhang ●[1,9] ✉, Hui Ding ●[2], Sonja Egert ●[3], Changhao Zhao ●[1], Lorenzo Villa[4], Lovro Fulanović ●[1], Pedro B. Groszewicz ●[5], Gerd Buntkowsky[3], Hans-Joachim Kleebe[6], Karsten Albe ●[4], Andreas Klein[7] & Jurij Koruza ●[1,8] ✉

Reversible field-induced phase transitions define antiferroelectric perovskite oxides and lay the foundation for high-energy storage density materials, required for future green technologies. However, promising new antiferro-electrics are hampered by transition´s irreversibility and low electrical resistivity. Here, we demonstrate an approach to overcome these problems by adjusting the local structure and defect chemistry, delivering NaNbO₃-based antiferro-electrics with well-defined double polarization loops. The attending reversible phase transition and structural changes at different length scales are probed by in situ high-energy X-ray diffraction, total scattering, transmission electron microcopy, and nuclear magnetic resonance spectroscopy. We show that the energy-storage density of the antiferroelectric compositions can be increased by an order of magnitude, while increasing the chemical disorder transforms the material to a relaxor state with a high energy efficiency of 90%. The results provide guidelines for efficient design of (anti-)ferroelectrics and open the way for the development of new material systems for a sustainable future.

The global goal to achieve net-zero CO₂ emissions by the middle of the century requires the development of efficient and environmentally benign technologies that can convert energy from renewable and sustainable resources such as sunlight and wind. Among the key components for the processing and distribution of renewable energy are efficient and reliable power electronics. Moreover, these are crucial for conversion and power control in hybrid or electric vehicles. Dielectric oxide capacitors[1,2] are characterized by high power densities of up to ~10⁸ W kg⁻¹, fast charge and discharge rates[3], high voltage and temperature endurance[4], as well as long lifecycles[5], which render them

particularly suitable for high-power and pulse-power technologies. The recoverable energy-storage density $W_{rec}$ of a dielectric is determined by the applied electric field $E$, and the induced polarization $P$,

$$W_{rec} = \int_{P_r}^{P_m} E\,dP, \qquad (1)$$

where $P_m$ and $P_r$ are the maximum and remanent polarization, respectively. Therefore, a combination of high $P_m$, low $P_r$, and a large applied field $E$, ensures the achievement of high energy-storage

[1]Non-metallic Inorganic Materials, Department of Materials and Earth Sciences, Technical University of Darmstadt, Darmstadt 64287, Germany. [2]Advanced Electron Microscopy, Department of Materials and Earth Sciences, Technical University of Darmstadt, Darmstadt 64287, Germany. [3]Eduard Zintl Institute for Inorganic and Physical Chemistry, Technical University of Darmstadt, Darmstadt 64287, Germany. [4]Materials Modeling Division, Department of Materials and Earth Sciences, Technical University of Darmstadt, Darmstadt 64287, Germany. [5]Department of Radiation Science and Technology, Delft University of Technology, 2600AA Delft, The Netherlands. [6]Institute of Applied Geosciences, Geomaterial Science, Technical University of Darmstadt, Darmstadt 64287, Germany. [7]Electronic Structure of Materials, Department of Materials and Earth Sciences, Technical University of Darmstadt, Darmstadt 64287, Germany. [8]Institute for Chemistry and Technology of Materials, Graz University of Technology, Graz 8010, Austria. [9]Present address: Department of Materials Science and Engineering, The Pennsylvania State University, University Park, PA 16802, USA. ✉e-mail: maohua.zhang.10@gmail.com; jurij.koruza@tugraz.at

densities. Besides high $W_{rec}$, a high energy efficiency ($\eta$) is also desired, which is defined as,

$$\eta = \frac{W_{rec}}{W_{rec} + W_{loss}} \times 100\%, \tag{2}$$

where $W_{loss}$ is the energy loss density and represents the area enclosed by the hysteresis loop.

Antiferroelectric perovskite oxides exhibit a phase transition from the nonpolar antiferroelectric (AFE) state to the polar ferroelectric (FE) state that can be triggered by the application of an electric field[6,7] and is characterized by a large field-induced $P_m$ and a near-zero $P_r$[8,9]. Therefore, AFE systems offer a much higher energy-storage density compared to ferroelectric and linear dielectric materials[10]. Unfortunately, the number of AFE oxides showing reversible AFE−FE phase transitions is very limited and most of them contain large amounts of toxic lead[11]. Due to the restriction of hazardous lead-containing substances in electrical and electronic products, the lead-free AFE material $NaNbO_3$ (NN) has been suggested as a possible alternative[12–14]. The FE order in $NaNbO_3$ is, however, readily stabilized by the application of an electric field and is maintained even after the field is removed[15–17]. Consequently, the transition is irreversible and double polarization hysteresis loops related to the reversible transition are absent, which hinders the realization of high energy-storage properties.

In the last decade, the development of new $NaNbO_3$-based AFE compositions for energy storage has been addressed by searching for new solid solutions that exhibit double polarization hysteresis loops at ambient conditions[18–20], and/or by elaborate chemical modifications to achieve high energy-storage density as a result of their relaxor-like behavior[12,21]. However, none of these $NaNbO_3$-based materials exhibited well-defined double polarization hysteresis loops at room temperature. Instead, the compositions with high energy-storage properties are characterized by slim hysteresis loops even at ultrahigh electric fields and are associated with relaxor-like behavior, with no experimental evidence of antiferroelectricity. In order to overcome the challenges in the design of $NaNbO_3$-based AFE materials, systematic understanding of the nature of the AFE − FE transition is needed, e.g., the delicate balance between the AFE and the FE order, the nucleation of the field-induced FE phase, the movement of the AFE−FE phase boundary during the phase transformation and its influence on the functional properties. Moreover, the role of defects and their migration are typically not considered in AFE materials exposed to high fields.

In this work, we designed a series of $NaNbO_3$-based materials with tailorable functional properties ranging from antiferroelectric to relaxor via judicious compositional modification and defect chemistry engineering. On the antiferroelectric side, well-defined double polarization hysteresis loops with minimal remanent polarization and 14 times higher energy-storage density than the prototypical $NaNbO_3$ are obtained, which are enabled by targeted modifications of the local structure and changes of the defect chemistry. Calculations within density-functional theory provide insights about the influence of intrinsic defects. The field-induced AFE−FE phase transition underlying the double loop is demonstrated by in situ high-energy synchrotron X-ray diffraction (XRD). The antiferroelectric−relaxor transition is featured with a significant increase in energy-storage efficiency from 30 to 90%. The structural changes from antiferroelectric to relaxor at different length scales are interpreted by pair distribution function (PDF) analysis and solid-state nuclear magnetic resonance (NMR) spectroscopy. On the relaxor side, the microstructural features and local chemical disorder characteristics are revealed by high-resolution transmission electron microscopy (HRTEM).

## Results and discussion

### Stability of antiferroelectric and relaxor states

Pure $NaNbO_3$ exhibits an irreversible AFE−FE transition[17,22], resulting in ferroelectric behavior upon repeated electric field (E-field) application, with a remanent polarization of 32.6 μC cm$^{-2}$ (Fig. 1a). Given the volatilization of alkali species during the high-temperature sintering process[23], alkali vacancies at the perovskite A site are inevitably created and are accompanied by the formation of holes $h^\bullet$ and thus p-type conductivity:

$$Na_{Na}^X \rightarrow V_{Na}' + h^\bullet + Na \uparrow \tag{3}$$

Our electronic structure calculations[24] show that the formation energy of a Na vacancy in $NaNbO_3$ is the lowest among all point defects (0.82 eV and 1.09 eV for $V_{Na}'$ and $V_O^{\bullet\bullet}$, respectively, calculated at processing conditions of 1360 °C and an oxygen partial pressure of 0.2 atm, where the defect equilibrium is established), with their concentration being about 2.2-times higher as the concentration of oxygen vacancies (Fig. 1i). Similar findings have been reported for $(K,Na)NbO_3$ bulk ceramics[25,26], single crystals[27,28], and thin films[29]. In addition, oxygen vacancies[30] and the multiple oxidation states of the B-site Nb ions[27] are expected to be present as additional point defects in these materials. While the dominant conduction mechanism has not been identified so far, these defects are expected to contribute to an increased dielectric loss in the low-frequency range (Fig. 1g).

Modification of $NaNbO_3$ with $SrSnO_3$ results in increased local chemical disorder and the stabilization of the antiferroelectric state[31]. The tolerance factor remains almost the same for NN (0.965) and $0.95NaNbO_3$–$0.05SrSnO_3$ (NN5SS, 0.964) samples (Supplementary Fig. 1), suggesting that it cannot be considered as the sole indicator of AFE order stabilization. $SrSnO_3$ addition enables room-temperature reversibility of the E-field-induced AFE−FE transition (Fig. 1b). However, the apparent remanent polarization is high (13.9 μC cm$^{-2}$) and the values are influenced by leakage current, which renders an ill-defined double polarization loop. We relate this behavior to the presence of an increased concentration of free charges, as detailed below. The incorporation of Sn into $NaNbO_3$ is difficult and, despite multiple calcinations, a minor amount of $SnO_2$ is observed (Supplementary Figs. 2 and 3). The excess Sr acts as a donor (Eq. (4)), as recently suggested in ref. [32] and theoretically in ref. [24]. The reaction can be written as:

$$Na_{Na}^X + SrSnO_3 \rightarrow Sr_{Na}^\bullet + e' + SnO_2(sec. phase) + 1/2 O_2 \uparrow + Na \uparrow \tag{4}$$

Although pure $NaNbO_3$ shows p-type conductivity, the $NaNbO_3$-$SrSnO_3$ solid solution exhibits n-type conductivity since excess Sr dominates over the Na vacancies[24]. This is directly reflected in the increased dielectric loss of NN5SS at low frequencies (Fig. 1g). Upon the application of large electric fields, the NN5SS sample transforms into the FE state at $E_{AFE-FE}$, whereby the moving phase boundary[15,33] and the changed domain state[34] facilitate a redistribution of free charges. These are likely to accumulate at grain boundaries, thereby forming a local electric field that partially stabilizes the induced FE state and is responsible for the large remanent polarization. Reducing the concentration of mobile charge carriers is thus recognized as one of the crucial issues for improving the AFE performance of $NaNbO_3$-based materials.

To this end, the compositions have been modified by adding $MnO_2$, which can act as an electron trap (Eqs. 5–7) and was previously reported to improve the resistance of other perovskites[35,36]; note that Mn prefers to have lower oxidation states in perovskites[37]. The exact valence state of Mn could not be determined, but the corresponding

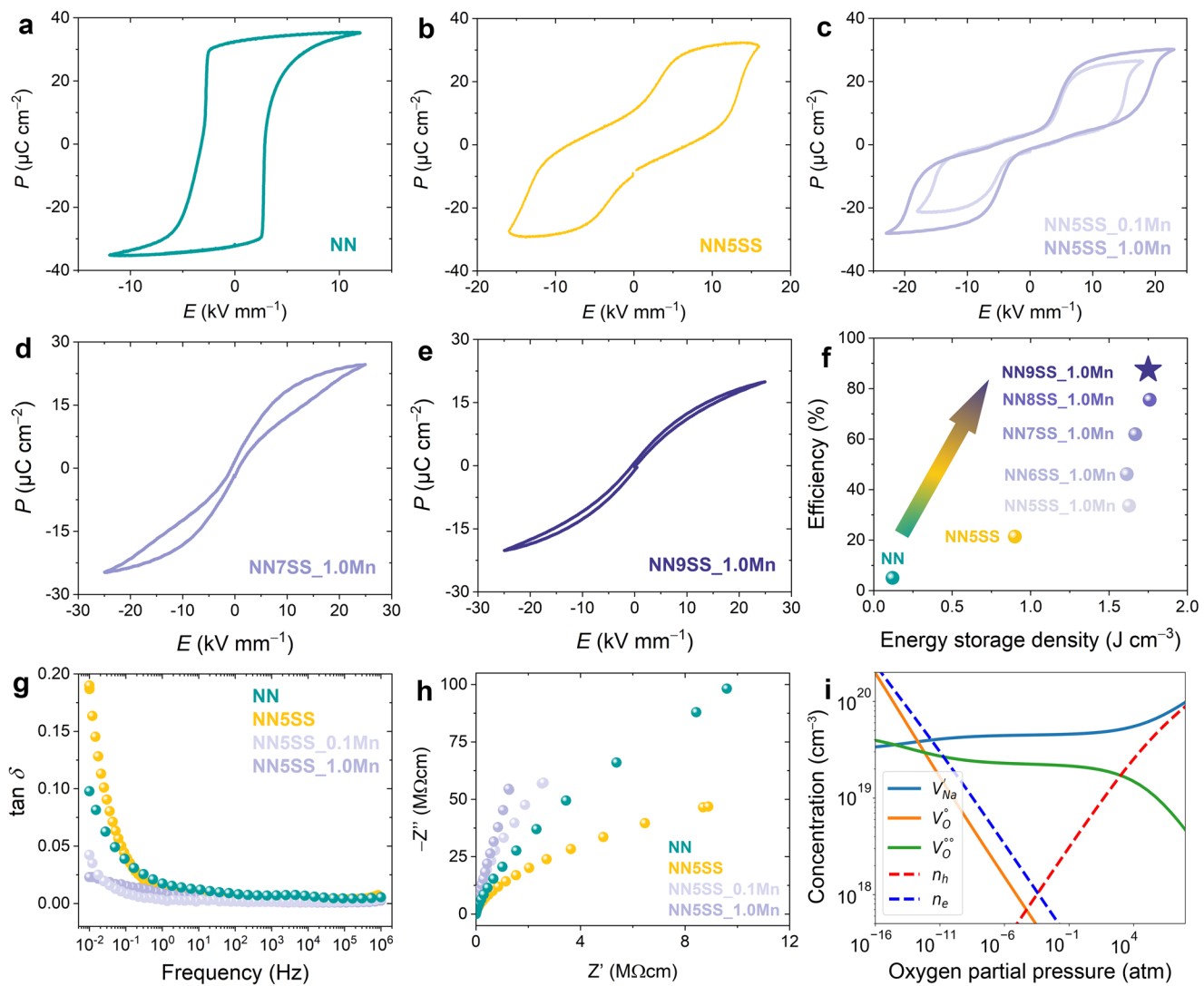

**Fig. 1 | Phase transition behavior and energy-storage performance of NaNbO$_3$-based antiferroelectrics and relaxors.** Polarization hysteresis loops of **a** NN, **b** NN5SS, **c** Mn-modified NN5SS (NN5SS_0.1Mn and NN5SS_1.0Mn), **d** NN7SS_1.0Mn, and **e** NN9SS_1.0Mn samples, obtained from the 2$^{nd}$ electric field cycle at 1 Hz. **f** Recoverable energy-storage density and efficiency properties of the investigated materials. **g** Frequency-dependent loss tangent and **h** Nyquist-plot of the NN, NN5SS, NN5SS_0.1Mn, and NN5SS_1.0Mn samples. **i** Concentrations of sodium vacancies ($V_{Na}''$), oxygen vacancies ($V_O^{\bullet\bullet}$ and $V_O^{\bullet}$), holes ($n_h$), and electrons ($n_e$) as a function of oxygen partial pressure, calculated at a temperature of 1360 °C.

acceptor reactions can be written as:

$$Mn_{Nb}^X + e' \rightarrow Mn_{Nb}' \tag{5}$$

$$Mn_{Nb}' + e' \rightarrow Mn_{Nb}'' \tag{6}$$

$$Mn_{Nb}'' + e' \rightarrow Mn_{Nb}''' \tag{7}$$

The addition of Mn to NN5SS successfully reduced the dielectric loss, in particular at the low-frequency range (Fig. 1g) and increased the resistance (Fig. 1h). As a result, the reduced amount of charge carriers had a positive effect on the double hysteresis loops (Fig. 1c). The samples exhibited a significantly lower remanent polarization of 3.2 μC cm$^{-2}$, corresponding to a tenfold decrease compared to the unmodified NN system. Note that this value is even lower as reported for comparable lead-based systems, such as (Pb,La)(Zr,Ti,Sn)O$_3$ (~4 μC cm$^{-2}$)[8]. The amount of Mn modification from 0.1 wt.% to 1.0 wt.% does not change the phase transition behavior, except that the critical field required to trigger the transition increases (Supplementary

Fig. 4). Due to the significantly reduced remanence, the energy-storage density increases from 0.12 J cm$^{-3}$ for NN and 0.90 J cm$^{-3}$ for NN5SS to about 1.70 J cm$^{-3}$ for the Mn-modified materials, which corresponds to a 14-fold increase. Moreover, the energy-storage density is stable over a wide temperature range from room temperature to 140 °C (max. measured temperature) due to the thermally-stable double loops (Supplementary Fig. 5).

Despite the increased energy-storage efficiency from 21% for NN5SS to 33% for NN5SS_1.0Mn, the values are still rather low. To further improve the efficiency, it is imperative to tune the shape of polarization loops from "square" to "slanted", i.e., to reduce the hysteresis and increase the distribution of critical transition fields[38]. To this end, we further increased the amount of SrSnO$_3$ substitution while maintaining the same MnO$_2$ content, namely NN100$x$SS_1.0Mn ($x$ = 0.06, 0.07, 0.08, and 0.09). The polarization responses of two selected compositions, NN7SS_1.0Mn and NN9SS_1.0Mn, are demonstrated in Fig. 1d, e, respectively. The hysteresis loops become slimmer with increasing SrSnO$_3$ (Supplementary Figs. 6 and 7) and hence, the energy-storage efficiency increases significantly from 33% for NN5SS_1.0Mn to about 90% for NN9SS_1.0Mn, as highlighted in Fig. 1f.

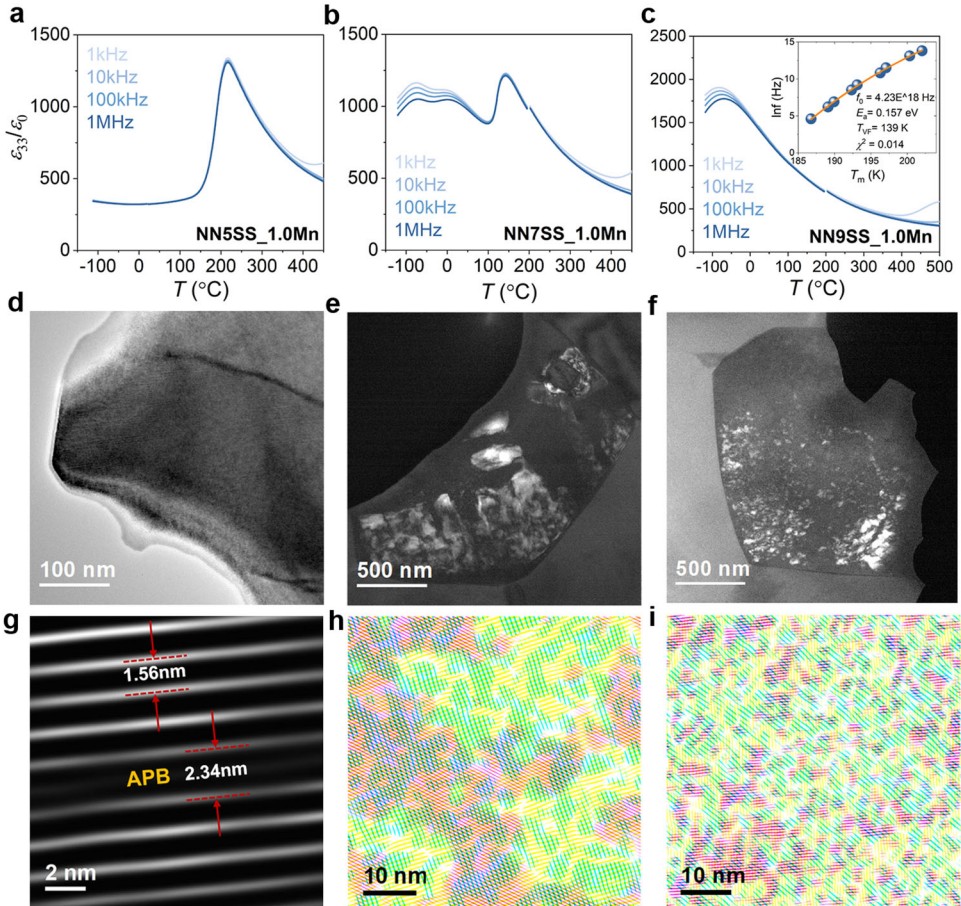

**Fig. 2 | Dielectric properties and TEM characterization of antiferroelectrics and relaxors.** Temperature-dependent dielectric permittivity of the **a** NN5SS_1.0Mn, **b** NN7SS_1.0Mn, and **c** NN9SS_1.0Mn samples, measured at a frequency of 1 kHz, 10 kHz, 100 kHz, and 1 MHz. The inset in (**c**) demonstrates the fitting of the frequency dispersion of the NN9SS_1.0Mn sample using the Vogel–Fulcher law. Domain morphology, obtained by the circled superlattice reflections of Fig. 3a–c in (**d**) bright-field (BF) and (**e, f**) corresponding centered dark-field (CDF) modes. **g–i** Inverse fast Fourier transform (IFFT) images of the HRTEM images of NN5SS_1.0Mn, NN7SS_1.0Mn, and NN9SS_1.0Mn, respectively. The original HRETM images can be found in Supplementary Fig. 8. The IFFT in (**g**) is obtained by masking the ¼(100) superlattice reflections, where regular lattice planes and the APB are highlighted with a modulation length of 1.56 nm and 2.34 nm, respectively. The IFFT images in (**h**) and (**i**) are a superposition of three IFFT images, obtained by masking the ½(10$\bar{1}$), ½(001) and ½(100) superlattice reflections, corresponding to the pink, green and yellow regions, respectively.

The structural origins of the observed changes in polarization loops and electrical behavior will be elaborated in the following.

## Dielectric and microstructural characterization

Given the possibility that relaxor behavior is the origin of the slanted hysteresis loop, we proceed with a thorough examination of the electrical and (micro)structural features. The NN5SS_1.0Mn sample with well-defined double loops (Fig. 1c) exhibits no dielectric frequency dispersion (Fig. 2a) and is characterized by homogeneous domain morphology (Fig. 2d) and uniform lattice fringes, interrupted by the antiphase boundaries (APBs) with darker line contrast[34] (Fig. 2g). These features are characteristic for antiferroelectric compositions. In contrast, frequency dispersion of the dielectric permittivity is observed for the NN9SS_1.0Mn sample (Fig. 2c), and its frequency-dependent dielectric permittivity can be well fitted using the Vogel–Fulcher relation (inset in Fig. 2c), a typical characteristic of relaxor ferroelectrics. The relaxor feature of the NN9SS_1.0Mn sample is further evidenced by randomly distributed polar nanoregions (PNRs) with a size of a few nm, as demonstrated by HRTEM images (Fig. 2f, i). Note that a direct characterization of local polarization in $NaNbO_3$-based materials is nontrivial, since both cations ($Na^+$ and $Nb^{5+}$) and anions ($O^{2-}$) are largely displaced[39], which is different from $PbZrO_3$-based materials where $Pb^{2+}$ displacements dominate. Nevertheless, the

polar nature of the nanoscale domains on the 1–20 Å scale will be evidenced by the complementary analysis of X-ray pair distribution functions (PDFs) below. The NN7SS_1.0Mn sample represents an intermediate composition between the AFE and relaxor states (Fig. 2b). It also exhibits inhomogeneous band-like domain structures that are perpendicular to each other (Fig. 2e), as well as nanometer-scale domains with a size of about 10–15 nm (Fig. 2h). Although the NN7SS_1.0Mn sample exhibits relaxor-like behavior, its frequency dependence of the dielectric permittivity cannot be fitted with the Vogel–Fulcher relation. These observations clearly indicate the onset of a relaxor state at higher $SrSnO_3$ content as the origin of the slimmer loops and higher energy-storage efficiency in the NN7SS-1.0Mn and NN9SS-1.0Mn samples.

## Average and local structures of antiferroelectrics and relaxors

Given the observation that the relaxor state is a critical factor for the enhanced energy-storage properties of NN7SS-1.0Mn and NN9SS-1.0Mn, it is relevant to understand what structural features are responsible for this behavior. The ¼ superlattice reflections in the selected area electron diffraction (SAED) pattern of the NN5SS_1.0Mn sample show that the local crystallographic structure exhibits the *Pbcm* space group (SG No. 57; Fig. 3a). This agrees well with the average *Pbcm* structure obtained from Rietveld refinement (Supplementary

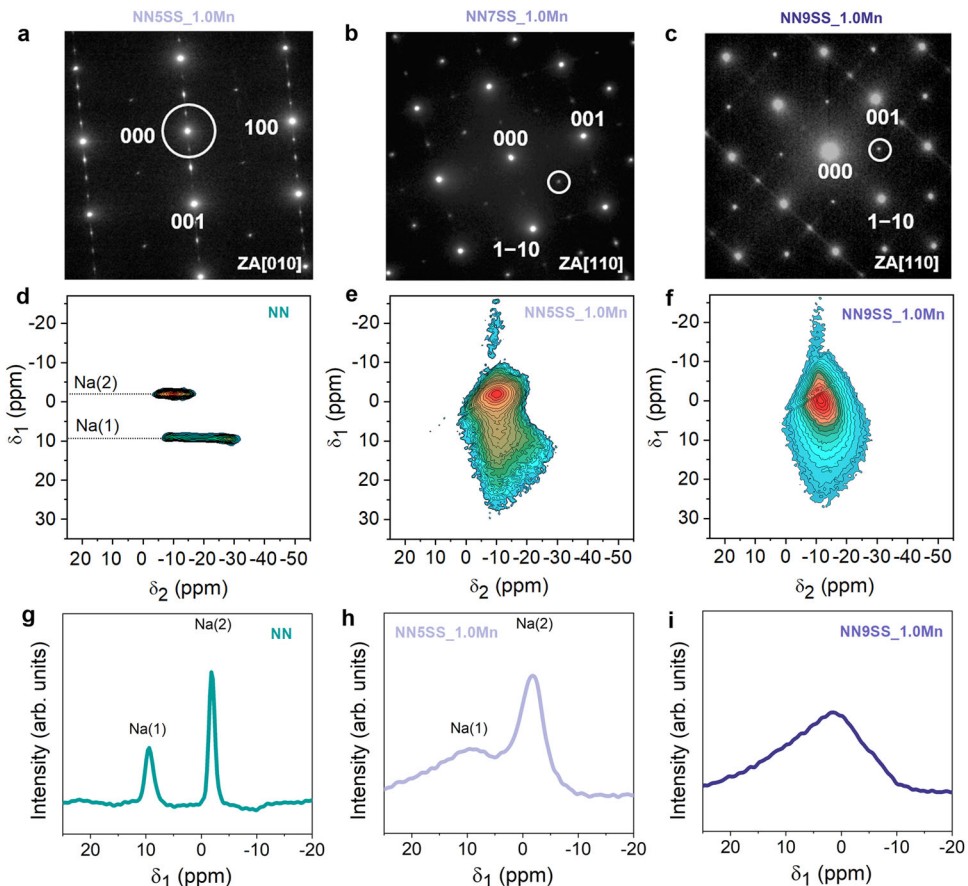

**Fig. 3 | Characterization of the local structures. a–c** The SAED patterns of the NN5SS_1.0Mn, NN7SS_1.0Mn, and NN9SS_1.0Mn samples recorded along the zone axis of the [010], [110], and [110] direction, respectively. The white circles highlight the selected reflections for the bright-field and centered dark-field images in

Fig. 2d–f. $^{23}$Na STMAS NMR spectra of **d** NN, **e** NN5SS_1.0Mn, and **f** NN9SS_1.0Mn samples. The signals of **g** NN, **h** NN5SS_1.0Mn, and **i** NN9SS_1.0Mn samples along the indirect dimension $\delta_1$.

Fig. 9). However, the ¼ reflections are not present in the NN7SS_1.0Mn and NN9SS_1.0Mn samples, as shown in Fig. 3b, c. A local high-resolution examination of the NN7SS_1.0Mn sample reveals a modulated structure, giving rise to distinct 1/6 superlattice reflections (Supplementary Fig. 10), which are characteristic of the *Pbnm* space group (SG No. 62, known as the high-temperature *R* phase in pure NaNbO$_3$[12,21,40]). However, such 1/6 superlattice reflections are observed only at a local scale, but are characterized by streaking features at a larger scale (Supplementary Fig. 10 and Fig. 3c) and therefore do not show long-range AFE order. The Rietveld refinement of the structure of NN9SS_1.0Mn shows that although the main reflections can be well fitted with the *Pbnm* model (Supplementary Fig. 11), the 1/6 super-lattice reflections are not satisfactorily described (Supplementary Fig. 12), which agrees with the argument that no long-range AFE order is present. Therefore, the NN9SS_1.0Mn sample is not considered AFE due to the absence of double loops and the observed relaxor behavior. Moreover, its structure differs from the *R* phase in pure NaNbO$_3$ (Supplementary Fig. 13), which is characterized by ordered super-lattice structures, and thus is referred to as *R'* phase in this study.

Further insight into the local structure can be gained by the application of solid-state nuclear magnetic resonance (NMR) spectroscopy[41]. The $^{23}$Na STMAS NMR spectra of NN and NN5SS_1.0Mn exhibit two distinct sodium sites consistent with the *Pbcm* space group, reflected by the observation of two well-resolved resonance peaks in the indirect dimension ($\delta_1$) centered at +9 ppm and −2 ppm (Fig. 3g, h). A summation of the signals of NN, NN5SS, NN5SS_0.1Mn, NN5SS_1.0Mn, and NN9SS_1.0Mn samples along the

indirect dimension $\delta_1$ is featured in Supplementary Fig. 14. In contrast, only one signal at an intermediate $\delta_1$ position is observed for the NN9SS_1.0Mn sample (Fig. 3i), hinting at a global phase transition. The position of this signal is comparable to the $^{23}$Na NMR spectrum of NaTaO$_3$[42], which exhibits the *Pbnm* space group at room temperature. This observation is thus consistent with the Rietveld refinement of the average structure and the TEM characterization of the local crystallographic structure of NN9SS_1.0Mn. However, the spectrum of NN9SS_1.0Mn reveals a highly disordered structure, with the line shape indicating significant distributions of local-structure-related NMR parameters, such as both quadrupolar coupling constant ($C_Q$) and isotropic chemical shift ($\delta_{iso}$). Statistical line shape analysis of the Na(1) signal of NN5SS_1.0Mn and the single signal of NN9SS_1.0Mn reveals featureless signal ridges, reflecting broad distributions of both $C_Q$ and $\delta_{iso}$ with an accompanying decrease of $C_Q$ (see Supplementary Fig. 15 and Supplementary Table 1). The addition of Mn in NN5SS_1.0Mn further reduces the distortion of NaO$_{12}$ cuboctahedra on the Na(1) site and introduces further Na−O bond length into the lattice; meanwhile NN9SS_1.0Mn is characterized by prominent Na−O disorder and a significantly lower average $C_Q$. The value of approximately 1300 kHz estimated from the line shape simulation is comparable to that of NaTaO$_3$ in the *Pbnm* structure; in addition, both $C_Q$ and $\delta_{iso}$ (−5.9 ppm) are comparable to the values of the Na(2) site in NaNbO$_3$ (−4.5 ppm, 1000 kHz)[42]. These are a characteristic trait of a disordered local structure of relaxor perovskite oxides[43,44], which provides further evidence that the composition is indeed in a relaxor state.

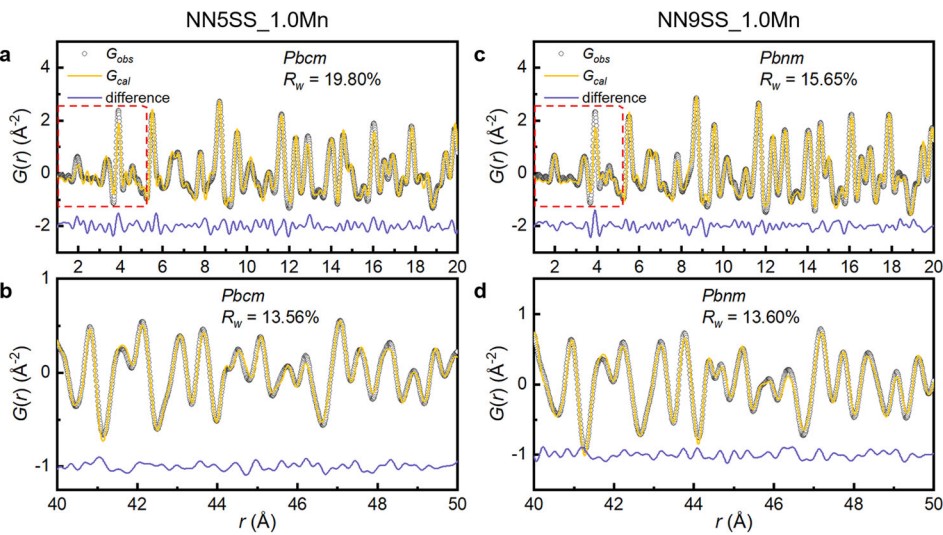

**Fig. 4 | PDF analysis of antiferroelectric and relaxor states.** X-ray pair distribution function $G(r)$ analysis of (**a**, **b**) NN5SS_1.0Mn and (**c**, **d**) NN9SS_1.0Mn samples in the $r$ range of $1-20$ Å and $40-50$ Å with the *Pbcm* and *Pbnm* space groups, respectively. The poor fit to the PDF below 5 Å is highlighted by dashed red squares.

### PDF characterization of antiferroelectric and relaxor states

To gain further insight into the evolution of the local structure from the antiferroelectric to the relaxor state, small-box modeling of the X-ray pair distribution functions (PDFs) of the NN5SS_1.0Mn and NN9SS_1.0Mn samples was performed with the *Pbcm* and *Pbnm* space groups, respectively, based on the obtained average long-range structures. The fitting results in the $r$ range of 1–20 Å, representing the short-range structure, and the 40–50 Å range, representing the intermediate-range structure, are depicted in Fig. 4. Overall, the used structural models agree with the experimental measurement. The fitting of the peaks in the 1–20 Å range is less satisfactory than that in the 40–50 Å range for the two samples, as indicated by a larger $R_w$ value in the first range. This is mostly caused by poor agreement in the small $r$ range (<6 Å) for both the NN5SS_1.0Mn and NN9SS_1.0Mn samples, as highlighted by the dashed squares. This could be related to the poor sensitivity of XRD for pair correlations involving oxygen, which are dominating this range. Note that the above $^{23}$Na NMR spectroscopy analysis provides information about the Na–O (2–3 Å) environments, while PDF from X-rays depicts the broader local environments. Another possible explanation is that the symmetry at the scale below 6 Å is instead *R3c* (SG No. 161), as suggested by high-resolution neutron scattering for pure NaNbO$_3$[39].

The orthorhombic lattice distortions ($b/a$, where $a$ and $b$ are orthorhombic lattice parameters) of the small-box models refined based on the PDFs of the two samples within the $r$ ranges of 1–20 Å and 40–50 Å, and those obtained from Rietveld refinement are listed in Table 1. The lattice distortion data of the NN5SS_1.0Mn sample at three different length scales are very similar, while the lattice distortion of the NN9SS_1.0Mn sample decreases with increasing length scale. In particular, the lattice distortion of 1.0108 at $r = 1-20$ Å is significantly larger than that of 1.0026 at $r = 40-50$ Å and that of 1.0014 from

Rietveld refinement, which represents the average long-range structure. A larger lattice distortion in the short range, as compared to the intermediate and the average structure, is a typical feature observed in relaxor ferroelectrics[45,46]. This indicates a local structural disorder in accordance with the NMR results, which results in a gradual averaging-out of the lattice distortion with increasing length scale. The observed strong diffuse scattering in the SAED patterns for the NN9SS_1.0Mn sample (Fig. 3c) and the parameter distributions observed in the NMR spectrum clearly indicate a deviation of the local structure from the average structure, which is due to the disordered local structure revealed by the PDF analysis.

### Field-dependent in situ structural analysis

Given the understanding of the local and average structures of the investigated AFE and relaxor materials, the structural change of the NN5SS_1.0Mn and NN9SS_1.0Mn samples under electric field was studied using in situ high-energy XRD to reveal the origin of the different functional responses. The NN5SS_1.0Mn composition is characterized by a field-induced phase transition from the AFE *Pbcm* symmetry[47] in the virgin state to the FE $P2_1ma$ symmetry[48], as evidenced by the reduced intensity of the ¼{843} superlattice reflection and the simultaneous appearance of the ½{312} superlattice reflection (Fig. 5a). These changes revert immediately after the *E*-field is removed, demonstrating the reversibility of the phase transition. Three stages can be identified according to the evolution of the $2\theta$ position of the {200} reflections, representing the pristine AFE state, field-induced FE state, and remanent AFE state, as shown in Fig. 5c. In previous studies of pure NN, only the pristine AFE and field-induced FE states were identified[22], which represent the structural origin of the irreversible phase transition. The presence of three different states is also confirmed by the calculated cell parameters, as shown in Fig. 5e, f. The cell parameters $a_{PC}$, $b_{PC}$, $\gamma$, and cell volume show a significant increase at the origin of the phase transition due to the change in the atomic displacements of the Na and Nb cations from antiparallel to parallel. Structural analysis reveals that the atomic displacements of the Na(1) ions relative to the symmetric centers in the cubic phase are 0.103 Å and 0.025 Å in the NaNbO$_3$ and NN5SS_1.0Mn samples (Supplementary Table 5), respectively. As a result, the orthorhombic lattice distortion decreases from 1.0113 for NaNbO$_3$ to 1.0085 for NN5SS_1.0Mn. This has two consequences: first, the critical field required to trigger the transition increases, as it becomes more difficult to break the centrosymmetric structure; second, the rigid structure is more likely to return the

### Table 1 | Calculated lattice distortion within the orthorhombic *ab* plane of NN5SS_1.0Mn and NN9SS_1.0Mn samples, as determined by analysis at different length scales

| | NN5SS_1.0Mn | NN9SS_1.0Mn |
|---|---|---|
| 1–20 Å | 1.0072 | 1.0108 |
| 40–50 Å | 1.0073 | 1.0026 |
| Rietveld refinement | 1.0087 | 1.0014 |

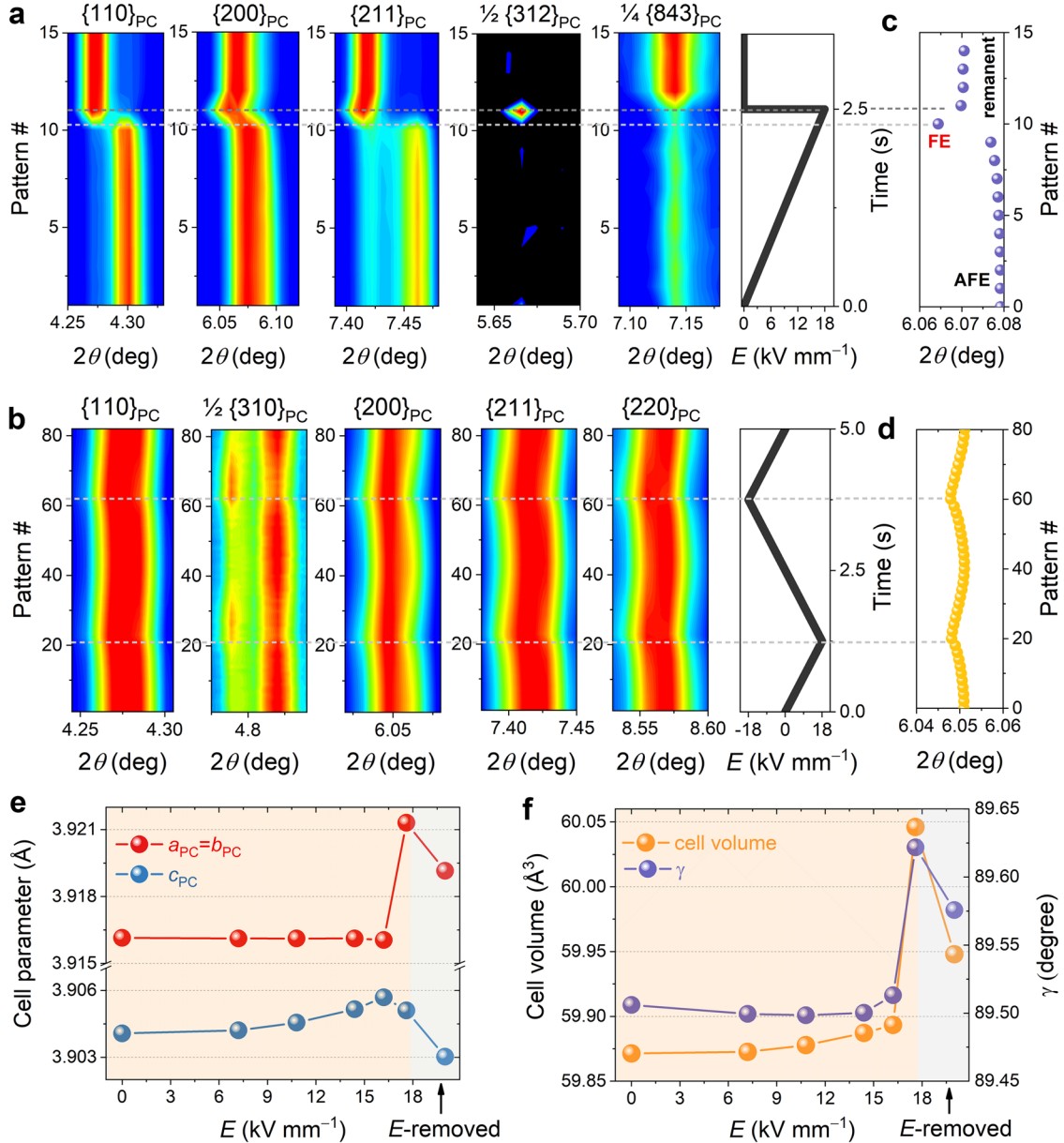

**Fig. 5 | In situ high-energy XRD characterization of antiferroelectric and relaxor states. a** Contour plots of representative primary and superlattice reflections of the NN5SS_1.0Mn sample under the application of a unipolar electric field of 18 kV mm⁻¹ and **b** the NN9SS_1.0Mn sample under the application of a bipolar electric field of 18 kV mm⁻¹. **c** Evolution of the {200} peak position for the

NN5SS_1.0Mn sample and **d** the NN9SS_1.0Mn sample. **e, f** Pseudocubic lattice parameters and primitive cell volume of the NN5SS_1.0Mn sample as a function of the electric field, obtained from LeBail fitting using a single-phase model. $a_{PC}$, $b_{PC}$, and $c_{PC}$ are the lattice parameters of the primitive cell, and $\gamma$ is the in-plane angle between the cell axes $a_{PC}$ and $b_{PC}$ (Supplementary Fig. 19).

field-induced structure to its original state and provide the restoring force for the reversible phase transition. It should be noted that the cell volume in the remanent state is slightly larger (0.12%) than that in the virgin state, while a remanent strain of 0.19% in the longitudinal direction is observed after the first electric field cycle (Supplementary Fig. 16). The difference in properties and structure between the first and subsequent cycles were also observed for PbZrO₃-based antiferroelectrics[49,50] (Supplementary Fig. 17), which can be attributed to the texturing of the sample caused by the application of an *E*-field. This is evidenced by the observation that the intensity of the ¼{843} superlattice reflection is stronger in the remanent state than in the virgin state.

In contrast, the contour plots of the NN9SS_1.0Mn composition (Fig. 5b) are characterized by continuous structural changes. The evolution of the {200} reflection (Fig. 5d) is due to the

electrostriction effect (Supplementary Fig. 18) and is clearly different from the abrupt 2θ change observed in the AFE composition (Fig. 5c). An intensity exchange is observed only in the ½{310} reflection associated with the tilting of oxygen octahedra[51], but not in the main reflections, e.g., the {200} reflection. Since electrostrictive strains and the tilting of the octahedra are intrinsically coupled[52,53], the intensity exchange is expected to result from the significant electrostrictive effect (Supplementary Fig. 18), where a large electrostriction coefficient $Q_{33} = 0.35$ m⁴ C⁻² and a large field-induced strain of 0.14% are observed (Supplementary Fig. 7). Therefore, no field-induced phase transition is observed for the NN9SS_1.0Mn composition under the present experimental conditions, which is due to the fact that it is a relaxor composition. These results confirm that the field-induced transition is not a necessary condition for high-energy-storage properties.

In summary, a series of newly developed lead-free energy-storage materials with tailorable functionalities from antiferroelectric to relaxor are presented. In situ high-energy XRD has ascertained that a reversible AFE−FE phase transition establishes the structural basis for the well-defined double polarization hysteresis loops with a low remanence, ten times smaller than that of pure NaNbO₃. This was enabled by targeted modifications of the local structure and changes of the defect chemistry. The addition of MnO₂ suppresses mobile charge carriers, strongly reduces the remanent polarization, and improves the resistance of materials. Increasing the chemical disorder by introducing more nonpolar SrSnO₃ into the antiferroelectric transforms the material into a relaxor state with a hysteresis-free polarization loop and a very high-energy-storage efficiency of 90%. The emergence of relaxor order is accompanied by the breakup of ordered superlattice structures of the antiferroelectric phase, the appearance of PNRs due to disorder in composition/microstructure, and hierarchical lattice distortion as a function of length scales. The suggested model for the interaction of charge carriers, phase boundaries, and the induced ferroelectric phase is generally applicable to other antiferroelectric materials and could help to improve their energy-storage performance. The above results were revealed and interpreted by a combination of XRD, TEM, PDF, and NMR techniques, which afford a complementary characterization at various length scales from a few angstroms to the average global structure. Such strategies provide guidelines for evaluating structure–property relationships and need to be considered for designing new generations of functional oxides.

## Methods

### Sample preparation

NaNbO₃-based samples, including NaNbO₃ (NN), 0.95NaNbO₃–0.05SrSnO₃ (NN5SS), 0.94NaNbO₃–0.06SrSnO₃ (NN6SS), 0.93NaNbO₃–0.07SrSnO₃ (NN7SS), 0.92NaNbO₃–0.08SrSnO₃ (NN8SS), 0.91NaNbO₃–0.09SrSnO₃ (NN9SS), and NN5SS, NN6SS, NN7SS, NN8SS, and NN9SS modified with different amounts of MnO₂ (0.1 wt%, 0.5 wt%, and 1.0 wt%), were prepared by solid-state reaction. The high-purity chemicals Na₂CO₃ (99.95%, Alfa Aesar, Germany), SrCO₃ (99.99%, Alfa Aesar, Germany), SnO₂ (99.90%, Alfa Aesar, Germany), and orthorhombic Nb₂O₅ (99.50%, Sinopharm, China) were dried at 200 °C for 8 h and then weighed in a stoichiometric ratio with 1 wt.% excess Na₂CO₃ to compensate for possible Na evaporation[54]. The chemicals were homogenized by planetary ball milling with yttria-stabilized zirconia balls in ethanol at 250 rpm for 12 h and then dried for 10 h. The dried powder mixtures were calcined at 850 °C for 4 h in alumina crucibles. The as-calcined powders were mixed with MnO₂ (99.90%, Alfa Aesar, Germany) and then ball milled at 250 rpm for 12 h, dried, and compacted into disks of 10 mm diameter and 1.5 mm thickness. The disks were subjected to cold isostatic pressing of 200 MPa before being sintered under various conditions with a packing powder of the same composition. The NN samples were sintered at 1355 °C for 3.5 h, while all NN100xSS (x = 0.05 − 0.09) samples with and without Mn modification were sintered at 1330 °C for 2 h.

### DFT

Density-functional Theory computations were performed using the Vienna ab initio simulation package (VASP)[55–57]. The exchange-correlation effects were treated using the Perdew–Burke–Ernzerhof[58] formalism of the generalized-gradient approximation (GGA). For the investigation of pure NaNbO₃, the electronic wave functions were described using the projector augmented-wave method[59,60], including valence and core states. The defect formation energies in all possible charge states were calculated, and the charge neutrality condition was solved self-consistently using the density of states obtained with DFT. This determined the Fermi level at which the defect charges are compensated by charge carriers. Once the Fermi level was known, the equilibrium defect concentrations and charge carrier concentrations were determined. The oxygen partial pressure dependence was obtained by linking it to the oxygen chemical potential (Supplementary Fig. 20). The phase of choice for all defect calculations is the orthorhombic Pbcm structure. DFT settings are consistent with the previous studies in ref. [24]. The supercell has 160 atoms, and its size was set to be 2 × 2 × 1. The plane wave basis set was expanded up to a 550 eV cutoff energy. The plane wave basis set was expanded up to a 550 eV cutoff energy. The Brillouin zone integration was performed using a Γ-centered 4 × 4 × 2 k-mesh for unit-cell calculations and a 2 × 2 × 2 k-mesh for supercell calculations. The Hellmann−Feynman forces criterion for atomic positions relaxation was set to 0.05 eV Å⁻¹ (see refs. [61,62]).

### Electrical characterization

The samples for the electrical measurements were cut and ground to a thickness of ~0.25 mm, sputtered with platinum to form symmetric electrodes that cover the entire surface of both larger sides, and then annealed at 400 °C to relieve mechanical stresses that may have developed during the cutting and grinding process. Polarization and strain hysteresis loops were obtained with a bipolar triangular wave with different amplitudes and frequencies using a modified Sawyer-Tower circuit. A high-voltage amplifier (20/20 C, Trek, USA) was employed as the high-voltage source. The temperature dependence of the dielectric permittivity and loss tangent were measured using an LCR meter (4192 A LF, Hewlett-Packard, USA). The data shown were recorded during the heating cycle at a rate of 2 °C min⁻¹, and the probe AC voltage was 1 V.

### TEM

Thin cross sections of the sintered NN5SS_1.0Mn, NN7SS_1.0Mn, and NN9SS_1.0Mn specimens were polished down to a thickness of 20 μm using a MultiPrep polishing system (Allied High-Tech Products Inc., USA) and diamond lapping films with different grain sizes ranging from 9 μm to 1 μm. The thin sections were then mounted on supporting molybdenum TEM grids (100 mesh; Plano, Germany) and Ar-ion milled to electron transparency using a DuoMill 600 (Gatan, USA). Subsequently, the TEM samples were lightly coated with carbon (Med 010, Liechtenstein) to minimize charging under the incident electron beam. TEM studies were performed using a JEM 2100 F microscope (JEOL, Tokyo, Japan) operating at 200 keV.

### NMR

The ²³Na solid-state NMR spectroscopic characterization was carried out at a Bruker Avance III spectrometer operating at a 7.1 T magnet with a carrier frequency of 79.38 MHz. For each composition, two pellets were cut to dimensions of 2.3 × 2.3 × 1.3 mm³, stacked on top of each other, and fitted to the middle section of a 4 mm zirconia rotor with TiO₂ as a packing powder. Spectra were recorded using a z-filtered and double-quantum filtered Satellite Transition Magic Angle Spinning (STMAS) sequence at a spinning rate of 10,000 ± 1 Hz[63]. Excitation, mixing, selective 90°, and selective 180° pulse lengths were 1.4, 2.1, 21.75, and 40.75 μs, respectively. In total, 2000 transients were averaged for each of the 110 increments with a relaxation delay of 1 s.

### Total scattering experiments

Atomic pair distribution functions (PDFs) were converted from total scattering functions, which were collected at the beamline P02.1 at Deutsches Elektronen-Synchrotron (DESY) with the same condition as given in the next section, except that the sample-to-detector distance was set as ~200 mm. Data was analyzed using the software PDFgetX3[64]. Each pattern was measured for 10 min to ensure a high signal-to-noise ratio at high scattering vector Q. Additional background pattern was

collected for 5 min without the sample; the signal which arises from the scattering of air, Kapton film, silicone oil, etc., was subtracted during the conversion of PDFs. Small-box PDF modeling was performed using the software PDFgui[65].

## High-energy diffraction measurement

The high-energy XRD data of NN5SS_1.0Mn sample were acquired at the DESY PETRA III P02.1 synchrotron beamline in transmission geometry with a Perkin-Elmer area detector (Perkin-Elmer, USA). The beam energy and spot size were 60 keV ($\lambda = 0.20727$ Å) and $0.25 \times 1.0$ mm$^2$, respectively. During the measurement, the samples were subjected to a unipolar electric field of 18 kV mm$^{-1}$ with a frequency of 0.2 Hz. The exposure time for each image was 250 ms. The two-dimensional XRD patterns were converted to one-dimensional patterns by integrating the intensity in different azimuthal regions using Fit2D[66]. LeBail fitting was performed using the GSAS program[67].

The synchrotron XRD data of NN9SS_1.0Mn sample were collected at the European Synchrotron Radiation Facility (ESRF) beamline ID15A in transmission geometry with a Pilatus 2 M CdTe area detector (Dectris Ltd., Switzerland) placed at a distance of 600 mm from the sample to collect diffraction data. The beam energy was 60 keV at a wavelength of 0.20664 Å and the spot size was set at $0.244 \times 0.244$ mm$^2$. During the measurement, a bipolar electric field of 18 kV mm$^{-1}$ with a frequency of 0.2 Hz was used. The exposure time for each image was 62.5 ms. The two-dimensional XRD patterns were converted to one-dimensional patterns by integrating the intensity in different azimuthal regions using DAWN[68].

## Data availability

The electrical properties, synchrotron, and NMR data generated in this study are provided in the Source Data file. More relevant data generated and/or analyzed during the current study are available from the first author and corresponding author on reasonable request.

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

## Acknowledgements

This work was supported by the Hessian State Ministry for Higher Education, Research and the Arts under the LOEWE collaborative project FLAME (Fermi level engineering of antiferroelectric materials for energy-storage and insulation systems). P.G. acknowledges financial support by the Dutch Research Council (NWO) for the ECCM Tenure Track funding under project number ECCM.006. We acknowledge DESY (Hamburg, Germany), a member of the Helmholtz Association HGF, for the provision of experimental facilities. Parts of this research were carried out at PETRA III using beamline P02.1. Beamtime was allocated for proposal I-20210563. We additionally acknowledge the European Synchrotron Radiation Facility (ESRF). Parts of this research was carried out at the beamline ID15 under proposal number MA 4993.

## Author contributions

M.Z. and J.K. conceived the idea of this work and designed the experiments. M.Z. prepared the materials under the supervision of J.K. and conducted all electrical measurements with the help of L.F. L.V. performed the DFT calculations under the supervision of K.A. A.K. contributed important insights into defect chemistry in discussions with J.K., M.Z., and L.V. D.H. is responsible for the TEM studies and the processing of the data under the supervision of H.-J.K. S.E. carried out the NMR measurements, processed the data, and received valuable feedback in discussion with P.G. and G.B. C.Z. conducted the total scattering and pair distribution function analysis. M.Z., C.Z., and J.K. conducted the high-energy XRD measurements, and M.Z. processed the structural data. M.Z. and J.K. drafted the first version of the manuscript and all authors participated in the writing and revision of the paper.

## Funding

## Competing interests

The authors declare no competing interests.
