## [Peer Review File · Nature Communications]

Tailoring high energy-storage NaNbO₃-based materials from antiferroelectric to relaxor statesREVIEWER COMMENTS

Reviewer #1 (Remarks to the Author):

(1) What are the noteworthy results?

Yes the results are noteworthy.

(2) Will the work be of significance to the field and related fields?

Definitely

(3) How does it compare to the established literature? If the work is not original, please provide relevant references.

The compositions have been modified by adding MnO₂, which can act as an electron trap and was previously reported to improve the resistance of other perovskites[35, 36]. So this indicates that the work is original.

Does the work support the conclusions and claims, or is additional evidence needed?

This work supports the conclusion and claims. However I have ask some questions to author that must be answer before publishing this research.

Are there any flaws in the data analysis, interpretation and conclusions? Do these prohibit publication or require revision?

Is the methodology sound? Does the work meet the expected standards in your field?

No, results are great and original.

Is there enough detail provided in the methods for the work to be reproduced?

Yes, the author have provided sufficient details for method. One can definitely reproduced the results.

Reviewer #2 (Remarks to the Author):

The manuscript by Zhang et al. reports a comprehensive study of phase tailoring of NaNbO₃ and its potential application as energy-storage dielectrics. A series of microscopic techniques are used to reveal a clear picture of the antiferroelectric and relaxor structures in modified NaNbO₃ ceramics. This work is surely of great interest to the dielectric realm and worthy of publication. Nevertheless, there are several issues in the current manuscript that need to be clarified before publication.

(1) The study of defects in NaNbO₃ is interesting. Please add the details of how the defect concentrations are calculated. Why is the calculation done at 1360 oC, where NaNbO₃ has become a cubic phase? It looks like that pure NaNbO₃ is p-type, then how does the excess

Sr degrade and then the MnO₂ improve the resistivity? It is better to add the corresponding defect-reaction equations.

(2) Can the stabilization of AFE phase in NN5SS be understood based on the tolerance factor, similar as that in PbZrO₃?

(3) For the NN9SS_1.0Mn that behaves like a relaxor, why the areas in Figure 2i (IFFT of some specific superlattice diffractions) represent PNRs? What is the R phase? It would be helpful to comparatively show the schematic lattice structures of NN5SS and NN9SS, to explain how and why the superlattice diffraction changes from 1/4 to 1/6. In ref. 12, the R phase is described still as an AFE phase, is it the same in this work for the relaxor-like NN9SS_1.0Mn?

(4) From the P-E loop (Figure 1c), NN5SS_1.0Mn shows a very small remnant polarization, so how should the large remnant in Figure 5e,f be understood? Besides, why the E-removed Cpc (Figure 5e) decreases compared with the pristine state?

Some minor issues:

(1) The figures are not arranged according to the order they appear in the text, which brings in confusion. Besides, the serial number of SI figures does not match that in the main text (e.g, Figure S8).

(2) Some denotations are not defined, e.g, the R phase, and the γ in Figure 5f.

Reviewer #3 (Remarks to the Author):

Reversible field-induced phase transitions define antiferroelectric perovskite oxides and lay the foundation for high energy-storage densities, required for future green technologies. This work demonstrates an approach of adjusting the local structure and defect chemistry to achieve well-defined double P-E loops and a relaxor state with a high energy efficiency of 90%. Various high-tech testing techniques were adopted to revealing the reversible phase transition and structural changes. However, some problems should be addressed.

The obtained well-defined P-E loop in NaNbO₃ is interesting and presents the novelty of this work. However, the author related the ill-defined double P-E loop and high remnant polarization to the high leakage current, due to the high concentration of free charges, which stabilize the induced FE state. The author added MnO₂ to act as an electron trap to reduce the amount of charge carriers and improve the resistance, thus produce double hysteresis loop. From this point, how about addition of excess Na element or adding other additives, they can also produce double hysteresis loop? The origin of double hysteresis loop does not clearly addressed yet. So, more solid evidence is required to support the point of view.

The author paid great efforts to produce a reversible AFE-FE phase transition and stated the AFE phase possess a Pbcm space group (NN5SS_1.0Mn), which should be a P phase.

However, it is found the slim P-E loops with increased energy storage density and efficiency are observed in NN9SS_1.0Mn, which possesses an R phase with a Pbnm space group.

These two compositions are of totally different phase structures. So, the question come out that what's the effect of the well-defined P-E loops in NN5SS_1.0Mn for the slim P-E loops in NN9SS_1.0Mn and the improved energy storage density and efficiency. It's more interesting to get a relaxor character in the P phase. From the viewpoint of energy storage performance, ultrahigh energy storage density with value over 10 J/cm³ was already reported in R phase in many NaNbO₃-based materials in the literatures, more than 5 times higher than the value of this work, and the mechanism of relaxor feature was also well-explained (Adv. Funct. Mater.2019, 29, 1903877). So, the concept is not new and the obtained energy storage

properties are poor. What's more, this manuscript even cannot induce an AFE-FE phase transition in NN9SS_1.0Mn, as said "no field-induced phase transition is observed for the NN9SS_1.0Mn composition under the present experimental conditions", in this case, high energy efficiency with low energy storage density is always observed. We even cannot evaluate the change of P-E loops and other electrical properties once the AFE-FE phase transition is induced, such as the hysteresis, remnant polarization, maximum polarization and the energy storage performance.

So, I feel this manuscript cannot meet the standard of Nature Communications.

Response Letter

Dear Editor,

Thank you very much for handling our manuscript. Here, we appreciate all the reviewers for their careful revision, constructive comments and the overall positive response. We have addressed all their comments below, carried out additional analysis, and have prepared a revised version of the manuscript. The respective changes in the manuscript are highlighted in yellow.

Reviewer #1: The results are great because here by modifying the composition of NN there is an increase in the energy-storage density up to 14-fold increase. Though there is an increase in energy-storage density, the author obtained the slanted polarization loop to further increase the efficiency by increasing the amount of SrSnO_3 while maintaining the content of MnO_2 same.

Reply: We thank the reviewer for her/his careful reading of the manuscript and helpful comments.

(1) How the quadrupole coupling C_Q and isotropic chemical shift effect the spectrum of NN(SS_1.0Mn? Meaning how quadrupole coupling and isotropic chemical shift behaves.

Reply: We thank the reviewer for their careful review of the NMR data. We assume that there is a typo in this comment and that the reviewer meant “NN9SS_1.0Mn”. The analysis of C_Q and δ_{iso} indeed provides valuable insights into the characteristics of the present sodium species. As depicted in Figure R1, a change of the apparent chemical shift in the indirect dimension (δ_2) of 1.7 ppm is observed between NN5SS_1.0Mn and NN9SS_1.0Mn, indicating a change of δ_{iso} and the quadrupolar induced shift, whereas the decrease of the 2nd order broadened line indicates a decrease of the quadrupolar coupling constant.

Figure R1 (Supplementary Fig 15). (a) Comparison of the projections of the Na(2) signals of NN5SS and NN5SS_1.0Mn and the total signal of NN9SS_1.0Mn onto the direct axis of the STMAS spectrum, revealing the characteristic quadrupolar broadened line shapes. Across the range of samples, the right flank becomes increasingly narrower and attenuated. (b) Line shape

simulation of NN5SS_1.0Mn with a set of 30 dependent lines and (c) line shape simulation of NN9SS_1.0Mn with a set of 40 dependent lines, yielding the parameters reported in Table R1.

Table. R1 (Supplementary Table. 1). Comparison of quadrupolar coupling constants C_Q , isotropic chemical shifts δ_{iso} , and distribution widths Δ thereof for the investigated materials. The data are obtained from line shape simulations as depicted in Figure R1 based on negatively correlated Gaussian distributions of both NMR parameters; values for NN_5SS have previously been reported in Zhang *et al.*, Chem. Mater. **2021**, 33, 266.

Composition	C_Q (kHz)	ΔC_Q (kHz)	δ_{iso} (ppm)	$\Delta\delta_{\text{iso}}$ (ppm)
NN5SS	1900	130	-2.8	1.2
NN5SS_1.0Mn	1600	160	-2.4	2.8
NN9SS_1.0Mn	1300	140	-5.9	3.2

In order to give a more quantitative answer, simulations of the quadrupolar broadened lines were carried out. The procedure employed by us involves generating projections of the Na(1) and Na(2) signal areas onto the direct axis of the ^{23}Na STMAS spectrum, then fitting the resulting pseudo-1D line shapes with positively correlated, but otherwise independent Gaussian distributions of both C_Q and δ_{iso} . In NN5SS, the Na(1) projection features a shoulder that, while significantly attenuated, indicates a line shape that is derived from the $\eta=0$, quadrupolar 2nd order broadened signal known for pristine NN (Figure R1a). Statistical line shape analysis can then demonstrate that the attenuation of this shoulder is related to a distribution of both C_Q and δ_{iso} with an accompanying decrease of C_Q (Zhang *et al.*, Chem. Mater. **2021**, 33, 266.).

For NN5SS_1.0Mn, the line shape is significantly less defined (Figure R1b). Qualitatively, its line shape can be understood as an extension of the NN5SS signal with an even more attenuated right shoulder. Indeed, a simulation assuming an analogous distribution of C_Q and δ_{iso} indicates that the line shape is consistent with an even lower C_Q and a broad distribution of δ_{iso} (Table. R1). Thus, the local environment of the Na(1) atoms is characterized by increased Na-O bond length disorder and decreased distortion of the NaO_{12} cuboctahedra through the addition of Mn. However, the accuracy of this kind of analysis decreases with decreasing definition of the pseudo-1D projections. Hence, the obtained NMR parameters should merely be viewed as support of the observed trends.

In NN9SS_1.0Mn, only one signal is discerned, which we attributed to the material assuming the *Pbnm* rather than the *Pbcm* structure. The total projection of the single signal yields an indirect projection that is narrower than that of NN5SS_1.0Mn while even more featureless. The described fitting procedure finds average values of 1300 kHz for C_Q and -5.9 ppm for δ_{iso} , with the distribution widths reported in Table R1.

The value for C_Q matches the value determined by Ashbrook *et al.* (Phys. Chem. Chem.

Phys., **2006**, 8, 3423) for NaTaO₃, an *Pbnm*-structure perovskite, indicating a similar degree of NaO₁₂ distortion, while δ_{iso} is shifted to a value more closely to the Na(2) signal of pristine NaNbO₃. The observed C_Q is similar to the value of 1000 kHz for Na(2) in pristine NN (Ashbrook *et al.*, Phys. Chem. Chem. Phys., **2006**, 8, 3423, Koruza *et al.* Acta Mater., **2017**, 126, 77, and Zhang *et al.*, Acta Mater., **2020**, 200, 127), indicating that the single broad signal in NN9SS_1.0Mn corresponds to a local environment more reminiscent of the Na(2) site. The distribution width of C_Q is comparable to that of NN5SS_1.0Mn, while the distribution width of δ_{iso} further increases, reflecting prominent Na-O bond length disorder with a comparable variation of the NaO₁₂ distortion.

It should be noted that differences in line width of the ²³Na NMR spectra can also be related to paramagnetic interactions with unpaired electrons from the Mn cations. (Bertmer *et al.*, Solid State Nucl. Magn. Reson., **2017**, 81, 1). However, the comparison of the spectra of NN5SS_1.0Mn and NN9SS_1.0Mn, both with the same nominal number of paramagnetic additives, supports our interpretation that the differences observed are mainly due to the incorporation of diamagnetic Sn species. Hence, contributions from paramagnetic interactions can be disregarded from the present analysis of distribution of ²³Na NMR parameters.

We have amended the manuscript on page 9 to include the following passages in order to reflect these additions and included Figure R1 and Table R1, together with an accompanying explanation, in the Supplementary Information.

“Statistical line shape analysis of the Na(1) signal of NN5SS_1.0Mn and the single signal of NN9SS_1.0Mn reveals featureless signal ridges, reflecting broad distributions of both C_Q and δ_{iso} with an accompanying decrease of C_Q (see Supplementary Fig. 15 and Supplementary Table. 1). The addition of Mn in NN5SS_1.0Mn further reduces the distortion of NaO₁₂ cuboctahedra on the Na(1) site and introduces further Na-O bond length into the lattice; meanwhile NN9SS_1.0Mn is characterized by prominent Na-O disorder and a significantly lower average C_Q . The value of approximately 1300 kHz estimated from the line shape simulation is comparable to that of NaTaO₃ in the *Pbnm* structure; in addition, both C_Q and δ_{iso} (−5.9 ppm) are comparable to the values of the Na(2) site in NaNbO₃ (−4.5 ppm, 1000 kHz)⁴¹.”

(2) One correction needs to be done on page 8: A summation of the signals of NN, NN5SS, NN5SS_0.1Mn, NN5SS_1.0Mn, and NN9SS_1.0Mn samples along the indirect dimension δ_1 is featured in Supplementary Fig. 9. I think here it should be Fig 8 not Fig 9.

Reply: Thank you for this careful observation; the figure number has been corrected.

(3) For NN5SS_1.0Mn and NN9SS_1.0Mn in Fig 4 (a) and (b), below 6 Ang, the structure is R3c due to poor fit to PDF. But let us say somehow someone is able to obtained good fit to PDF.

In that case, do in both cases, still below 6 Å, the structure will be *R3c* or not? Also, apart from fitting to PDF, is there any other ways to predict that the structure is *R3c*.

Reply: Thank you for pointing out this important question. The structures of antiferroelectrics are interesting but also very complicated, e.g., even the ground state structure and polar/antipolar nature of prototypical PbZrO_3 is still under debate. Similarly, the local structure of NaNbO_3 is also controversial. Although early seminal work by Megaw *et al.* suggested the orthorhombic *Pbcm* symmetry (Ferroelectrics, **1973**, 7, 87), more recent high-resolution neutron scattering data from Jiang *et al.* (Phy. Rev. B, **2013**, 88, 014105) revealed that the short-range structure (≤ 5 Å) cannot be satisfactorily described by the *Pbcm* space group, as shown in Figure R2. The above reasons justify the use of complementary characterization techniques in our study in order to draw a solid conclusion. If, as the reviewer assumed, the PDF data for NN5SS_1.0Mn were well fitted with the *Pbcm* structure, it is very likely that the structure, including that below 6 Å, is *Pbcm* and not *R3c*. This is because the presence of the *Pbcm* structure is supported by a combination of two local structural characterizations (PDF and NMR) and an average structural characterization (Rietveld refinement).

However, the poor fit to the PDF data below 6 Å raises the question of whether the *Pbcm* (or *Pbnm*) model can well describe the structures of the NN5SS_1.0Mn (NN9SS_1.0Mn) samples at all length scales, which we cannot simply ignore. Note that although both NMR and PDF characterize local structures, they represent different aspects and are therefore considered complementary. ^{23}Na NMR spectroscopy mainly provides information about the Na-O environment, while PDF from X-rays depicts the broader local environments, which cannot fully clarify the uncertainty. As discussed in the manuscript, the range below 6 Å is strongly dominated by pair correlations involving O, for which X-rays are not sensitive enough due to the poor scattering of oxygen (compared to Nb). Therefore, the contribution of Na-O distances (2-3 Å) is highly overshadowed by Na-Na (just below 4 Å), Nb-O (at 2 Å), and Nb-Nb (at 4 Å). For the above reasons, we decided to keep the question about the local structure open.

Density functional theory (DFT) calculation is an alternative approach for predicting the structure, which has shown that the ground state of NaNbO_3 is the *R3c* structure with a smaller free energy than that of the *Pbcm* structure (Shimizu *et al.* Dalton Trans., **2015**). However, these calculations consider temperatures of 0 K, therefore their extrapolation to room temperature structures should be considered with care.

In summary, we regret that existing experimental techniques and our data cannot provide a more accurate answer regarding the presence of the *R3c*. In the future, we hope to conduct a more detailed analysis using neutron diffraction, which could possibly provide further evidence. One sentence is added to the corresponding part on page 11 in the revised manuscript:

“Note that the above ^{23}Na NMR spectroscopy analysis provides information about the Na-O (2-3 Å) environment, while PDF from X-rays depicts the broader local environments.”

Figure R2. Neutron pair distribution function $G(r)$ analysis of NaNbO_3 at 490 K with the (a) $Pbcm$ and (b) $R3c$ space groups (Jiang *et al.* *Phy. Rev. B*, **2013**, 88, 014105).

(4) For NN5SS_1.0Mn and NN9SS_1.0Mn in Fig 5, the strength of the electric field is 18 kV/mm. Now for NN9SS_1.0Mn, one cannot see the field induced phase transition. Is it possible to increase the electric field strength to induced the phase transition? Also, from Figure 5 (d), is it true that one has only AFE state for NN9SS_1.0Mn?

Reply: First, we would like to answer the second part of this question. At zero field, the NN9SS_1.0Mn sample exhibits a relaxor state rather than an AFE phase. We made this conclusion based on a series of complementary characterizations, including macroscopic electrical measurements (P - E loops in **Figure 1e** and permittivity in **Figure 2c**), as well as structural microscopic measurements (HRTEM in **Figure 2f, i** & **Figure 3c** and PDF in **Figure 4** & **Table 1**). Due to the absence of the AFE phase in the NN9SS_1.0Mn sample at zero-field conditions, it is therefore not possible to induce an AFE-FE phase transition when an E -field is applied. An additional confirmation for the absence of field-induced transitions and presence of a relaxor state can be obtained if the measured strain is plotted against the square of the polarization (Figure R3). Here, the clearly linear response demonstrates that the strain in **Figure 5d** is contributed only by electrostriction, while no domain contributions or phase transitions are observed. Figure R3, together with the field-induced strain hysteresis loops of the investigated materials are now included in the Supporting Information as Figure S18 and Figure S19, respectively.

Figure R3 (Supplementary Fig 18). The strain-polarization² ($S\text{-Sign}(P)\cdot P^2$) loop of the NN9SS_1.0Mn sample. The fitted electrostrictive coefficient ($0.35\text{ m}^4/\text{C}^2$) is highlighted.

One might think that the NN9SS_1.0Mn sample is in the AFE state based on the observation that its main XRD reflections correspond to the $Pbnm$ model, which is considered to be AFE in pure NaNbO_3 . However, they are not AFE for the following reasons:

- a) No double loops, characteristic of a field-induced phase transition, have been reported in the $Pbnm$ structure, which does not seem to be AFE in NN9SS_1.0Mn. Therefore, despite the presence of antiferrodistortion, it cannot be considered AFE since the transition cannot be triggered. For this reason, we have decided to call it the R' phase.
- b) The $1/6$ superlattice reflections can be observed in the SAED patterns only at a very local scale (Supplementary Fig. 10). However, such $1/6$ superlattice reflections are less visible when taken from a large region (**Figure 3b & 3c**). This is due to the random distribution of nanodomains in different orientations, which leads to the streaking feature in the electron diffraction patterns. That is, NN9SS_1.0Mn sample does not have long-range AFE order that would otherwise lead to the well-defined $1/6$ superlattice reflections, as confirmed by the high amount of structural order in the NMR results.

We modified the text in the corresponding section on page 8 of the manuscript, which refers to both the NN7SS_1.0Mn and NN9SS_1.0Mn samples, to provide a better understanding of the behavior of the NN9SS_1.0Mn sample:

“A local high-resolution examination of the NN7SS_1.0Mn sample reveals a modulated structure, giving rise to distinct $1/6$ superlattice reflections (Supplementary Fig. 10), which are

characteristic of the *Pbnm* space group (SG No. 62, known as the high-temperature *R* phase in pure NaNbO_3 ^{12, 21, 39}). However, such $1/6$ superlattice reflections are observed only at a local scale, but are characterized by streaking features at a larger scale (Supplementary Fig. 10 and **Figure 3c**) and therefore do not show long-range AFE order. The Rietveld refinement of the structure of NN9SS_1.0Mn shows that although the main reflections can be well fitted with the *Pbnm* model (Supplementary Fig. 11), the $1/6$ superlattice reflections are not satisfactorily described (Supplementary Fig. 12), which agrees with the argument that no long-range AFE order is present. The NN9SS_1.0Mn sample is not considered AFE due to the absence of double loops and the observed relaxor behavior. Moreover, its structure differs from the *R* phase in pure NaNbO_3 (Supplementary Fig. 13), which is characterized by ordered superlattice structures, and thus is referred to as *R'* phase in this study.”

Furthermore, the reviewer suggested to increase the applied *E*-field beyond 18 kV/mm in the synchrotron measurement in **Figure 5d**. Please note that measurements at higher fields have been conducted in **Figure 1e**, where the sample was exposed to a bipolar field of 25 kV/mm, albeit without simultaneous X-ray measurement (also here, no field-induced transition has been observed). Application of such high electric fields during in situ synchrotron measurements is unfortunately not possible, mostly for two reasons:

- a) The samples for synchrotron measurements have to be thicker (0.3-0.5 mm), as compared to the samples for electrical measurements (0.2-0.3 mm), due to the limited beam size and detector sensitivity. It is well known that the dielectric breakdown strength decreases with increasing the sample's thickness (Neusel *et al.* J. Eur. Ceram. Soc., **2015**, 35, 113), therefore the thicker synchrotron-measurement samples had a lower breakdown strength and could only be exposed to fields of up to 18 kV/mm.
- b) The incoming high-energy X-rays are inducing a minor radiation damage during the measurement (likely due to absorption and photoelectron generation); while this influence is typically small, it can be amplified at higher electric fields and additionally decrease the dielectric breakdown strength (Hughes, Radiat. Eff., **1975**, 26, 225-235, Jain and Nowick, J. Appl. Phys., **1982**, 53, 485).

The corresponding sections on page 13 in the revised manuscript is revised as follows:

“In contrast, the contour plots of the NN9SS_1.0Mn composition (**Figure 5b**) are characterized by continuous structural changes, mainly due to intrinsic lattice response. An intensity exchange is observed in the $1/2310$ reflection, which hints at a small contribution from domain switching. The evolution of the 200 reflection (**Figure 5d**) is due to the electrostriction effect

(Supplementary Fig. 18) and is clearly different from the abrupt 2θ change observed in the AFE composition (Figure 5c). An intensity exchange is observed only in the $\frac{1}{2}310$ reflection associated with the tilting of oxygen octahedra⁵⁰, but not in the main reflections, e.g., the 200 reflection. Since electrostrictive strains and the tilting of the octahedra are intrinsically coupled^{51, 52}, the intensity exchange is expected to result from the significant electrostrictive effect (Supplementary Figs. S18), where a large electrostriction coefficient $Q_{33} = 0.35 \text{ m}^4 \text{ C}^{-2}$ and a large field-induced strain of 0.14% are observed (Supplementary Figs. S7). Therefore, no field-induced AFE-FE phase transition is observed for the NN9SS_1.0Mn composition under the present experimental conditions, which is due to the fact that it is a relaxor composition. These results confirm that the field-induced transition is not a necessarily condition for high energy storage properties.”

(5) Why bipolar electric field is used for NN9SS_1.0Mn?

Reply: We measured the NN9SS_1.0Mn sample using both unipolar and bipolar electric fields. Since a unipolar cycle is half of a bipolar cycle, we decided to show here the bipolar electric field for NN9SS_1.0Mn. We actually intended to show a bipolar cycle for NN5SS_1.0Mn as well, but as described in our response to the previous comment, *in situ* electric field synchrotron measurements for lead-free antiferroelectrics are challenging due to the combination of large thickness, high electric fields, and radiation conditions. Since breakdown for the NN5SS_1.0Mn sample occurred after the antiferroelectric–ferroelectric phase transition, we could only record its response in a unipolar cycle. Please note that the main reason for conducting these measurements was to evaluate the *E*-field induced transition, which takes place during the first half of the unipolar cycle (first quarter of the bipolar cycle).

(6) What is the criteria to select the length scale? For example, the author has used length scale up to 50 Ang (from Figure 4).

Reply: This is a very good question. The PDF analysis in this study aims to provide further insight into the local structures of the materials studied at different length scales. The unit cell of the NN5SS_1.0Mn sample is about $5 \times 5 \times 15 \text{ \AA}^3$, while that of the NN9SS_1.0Mn sample is $5 \times 5 \times 25 \text{ \AA}^3$ (Figure R6). At the smallest scale, an *r* range of less than 10 Å characterizes the short-range local structure within a primitive perovskite cell. On the other hand, the structural information obtained by Rietveld refinement shows the long-range average structure of the whole material. A length scale of 50 Å corresponds approximately to 2-3 times the size of the unit cell and describes the structure at intermediate length scales bridging the two extreme cases. Therefore, the choice of 50 Å as the maximum *r*-range is reasonable, which is also

commonly used in the literature (Jiang *et al.* Chem. Mater. **2017**, 29, 10, 4244 and Hou *et al.* J. Eur. Ceram. Soc., **2018**, 38, 971). Note that the maximum allowable r in the PDF is limited by the Q -resolution (Egami and Billinge, **2012**, Underneath the Bragg peaks: structural analysis of complex materials. Newnes). In this case, the peak amplitude falls off above 50 Å due to the finite Q -resolution of the measurement, resulting in a large error in the refined parameters.

(7) The two sample NN5SS_1.0Mn and NN9SS_1.0Mn at room temperature has $Pbcm$ and $Pbnm$ symmetry. The Table 1 of supplementary material shows the lattice parameters for $Pbcm$. As one understand that we get phase transition for NN5SS_1.0Mn case. Is it possible for author to supply the lattice parameters for NN9SS_1.0Mn sample, if possible?

Reply: We thank the reviewer for this suggestion. The lattice parameters of NN9SS_1.0Mn have been included in the Supplementary Information (Supplementary Table 2). In addition, the refined structural parameters, including atomic positions, are also included (Supplementary Table. 3, 4, 5, & 6).

Supplementary Table 2. The refined cell parameters of NN, NN5SS, NN5SS_0.1Mn, NN5SS_0.5Mn, NN5SS_1.0Mn, and NN9SS_1.0Mn samples.

	a (Å)	b (Å)	c (Å)	V (Å ³)	Space Group	R_w
NN	5.50346 (10)	5.56652 (10)	15.53871 (24)	476.031 (10)	$Pbcm$	0.0524
NN5SS	5.51535 (7)	5.56344 (6)	15.61913 (21)	479.263 (7)	$Pbcm$	0.0892
NN5SS_0.1Mn	5.51512 (12)	5.56325 (11)	15.6168 (4)	479.154 (11)	$Pbcm$	0.0623
NN5SS_0.5Mn	5.51441 (12)	5.56248 (11)	15.6166 (4)	479.021 (11)	$Pbcm$	0.0622
NN5SS_1.0Mn	5.51354 (13)	5.56140 (12)	15.6151 (4)	478.805 (12)	$Pbcm$	0.0665
NN9SS_1.0Mn	5.53661 (16)	5.54415 (12)	23.4711 (5)	720.465 (9)	$Pbnm$	0.0628

Supplementary Table 6. Refined structural parameters of NN9SS_1.0Mn with *Pbnm* model.

	x (Å)	y (Å)	z (Å)	U_{iso}
A1-site	0.5071(9)	0.5020(0)	0.25	0.0129(7)
A2-site	0.9984(6)	0.0018(7)	0.0836(5)	0.0129(7)
B1-site	0.5	0	0	0.0044(6)
B2-site	0.4983(2)	0.0237(7)	0.3338(1)	0.0044(6)
O1-site	0.5457(5)	0.4870(0)	0.25	0.0191(5)
O2-site	0.0601(5)	0.2417(1)	0.0889(5)	0.0339(9)
O3-site	0.7393(4)	0.3005(1)	0.0087(2)	0.0279(9)
O4-site	0.7170(1)	0.3413(6)	0.3413(6)	0.0057(4)
O5-site	0.7115(5)	0.2832(9)	0.6700(0)	0.0072(6)
R_{wp}			6.28	
GOF (χ^2)			2.89	

Reviewer #2: The manuscript by Zhang *et al.* reports a comprehensive study of phase tailoring of NaNbO₃ and its potential application as energy-storage dielectrics. A series of microscopic techniques are used to reveal a clear picture of the antiferroelectric and relaxor structures in modified NaNbO₃ ceramics. This work is surely of great interest to the dielectric realm and worthy of publication. Nevertheless, there are several issues in the current manuscript that need to be clarified before publication.

Reply: We thank the reviewer for the interest and constructive comments that definitely improve this paper.

(1) The study of defects in NaNbO₃ is interesting. Please add the details of how the defect concentrations are calculated. Why is the calculation done at 1360 °C, where NaNbO₃ has become a cubic phase? It looks like that pure NaNbO₃ is p-type, then how does the excess Sr degrade and then the MnO₂ improve the resistivity? It is better to add the corresponding defect-reaction equations.

Reply: Thanks for the question. First, the defect formation energies in all possible charge states are calculated in dependence on Fermi energy. Using the density of states obtained with DFT, we solve the charge neutrality condition self-consistently to determine the Fermi level at which the defect charges are compensated by charge carriers. Once we know the Fermi level, we know the equilibrium defect concentrations and the charge carrier concentrations. The dependence on the oxygen partial pressure is obtained by linking it to the oxygen chemical potential (Figure R4). For more details, we refer to a more comprehensive work on defect thermodynamics in NaNbO₃ (Villa and Albe, Phys. Rev. B, **2022**, 106, 134101). The above details are now included in the revised manuscript on page 16:

“First, the defect formation energies in all possible charge states were calculated. Using the density of states obtained with DFT, we solved the charge neutrality condition self-consistently to determine the Fermi level at which the defect charges are compensated by charge carriers. Once we know the Fermi level, we know the equilibrium defect concentrations and the charge carrier concentrations. The dependence on the oxygen partial pressure is obtained by linking it to the oxygen chemical potential (Supplementary Fig. 20).”

Figure R4 (Supplementary Fig. 20). Fermi level as a function of the oxygen partial pressure.

The calculation was performed at 1360 °C, since this is close to the sintering temperature of these samples. Although this is indeed the cubic phase region, as rightfully stated by the reviewer, we note that the defect equilibrium is in fact established at high temperature and the phase transition does not influence the defect concentration. Albeit the defect concentration may indeed change during cooling, the dominant carrier type remains the same at lower temperatures. To double-check this statement, we additionally carried out a calculation for 727 °C (1000 K) and found that the ratio of sodium vacancy to oxygen vacancy concentration was 2.3, which is very similar to the ratio of 2.2 calculated for 1360 °C (1628 K). The calculated results are thus qualitatively identical.

As correctly pointed out by the reviewer, pure NaNbO₃ is *p* type. The calculation of the equilibrium electron chemical potential (Villa *et al.*, *J. Appl. Phys.*, **2022**, 131, 124106) shows that the formation energies of Na vacancies are the lowest in energy in Na-poor/O-rich conditions. As known from the literature, alkaline species are characterized by high volatilization during the high-temperature sintering process (Lamoreaux *et al.* *J. Phys. Chem. Ref. Data*, **1984**, 13, 151). The corresponding defect-reaction equation can be written as:

In the NaNbO₃-SrSnO₃ solid solution, excess Sr is present due to difficult incorporation of Sn and thus Sr acts as a donor and induces *n*-type conductivity due to the deficiency of Sn:

In this case, the NaNbO₃-SrSnO₃ solid solution is *n* type, since the concentration of $\text{Sr}_{\text{Na}}^{\bullet}$ is higher than that of V'_{Na} and thus $\text{Sr}_{\text{Na}}^{\bullet}$ dominates (Villa *et al.*, *J. Appl. Phys.*, **2022**, 131, 124106). This is directly reflected in the increased dielectric loss of NN5SS at low frequencies

(Figure 1g). The subsequently added MnO₂ acts as an acceptor and is expected to reduce the electron concentration in the solid solution:

However, please note that the exact valence states of Mn could not be determined. A more detailed analysis of these at different oxygen partial pressures and using advanced spectroscopic methods is planned for the future.

The above discussion, as well as the corresponding defect reactions, have been included into the main manuscript on Pages 3-5 (Equations 3-7).

(2) Can the stabilization of AFE phase in NN5SS be understood based on the tolerance factor, similar as that in PbZrO₃?

Reply: We agree with the reviewer that the tolerance factor is an important indicator, albeit it is not the only one and cannot be used as a simple predictor. For example, the tolerance factor of pure NaNbO₃ is 0.965 and remains almost unchanged for NN5SS (0.964). Despite this, the stability of the AFE phase was found to markedly increase when the Sr²⁺ (1.44 Å, CN=12) and Sn⁴⁺ (0.69 Å, CN=6) ions partially substitute the Na⁺ (1.39 Å, CN=12) and Nb⁵⁺ (0.64 Å, CN=6) ions, respectively. We note that the tolerance factor of NN5SS is higher than for some other typical antiferroelectrics, like PbZrO₃ (0.964), AgNbO₃ (0.929), and some other representative NaNbO₃-based systems (Figure R5). In our opinion, a suitable tolerance factor is an important precondition for possible stability of AFE phase; however, based on our investigations using NMR, the disorder of the local structure plays the crucial role (for detailed discussion, see Zhang *et al.*, Chem. Mater., **2021**, 33, 266).

One sentence was added to the corresponding section in the revised manuscript:

“The tolerance factor remains almost the same for NN (0.965) and NN5SS (0.964) samples (Supplementary Fig. 1), suggesting that it cannot be considered as the sole indicator of AFE order stabilization.”

Figure R5 (Supplementary Fig. 1). Tolerance factor of the NaNbO_3 - SrSnO_3 solid solution and other representative systems. The ionic radii data are taken from the Shannon Database of Ionic Radii (<http://abulafia.mt.ic.ac.uk/shannon/ptable.php>).

(3) For the NN9SS_1.0Mn that behaves like a relaxor, why the areas in Figure 2i (IFFT of some specific superlattice diffractions) represent PNRs? What is the R phase? It would be helpful to comparatively show the schematic lattice structures of NN5SS and NN9SS, to explain how and why the superlattice diffraction changes from 1/4 to 1/6. In ref. 12, the R phase is described still as an AFE phase, is it the same in this work for the relaxor-like NN9SS_1.0Mn?

Reply: We thank the reviewer for raising these important questions and giving us a good suggestion. For the first question, we used the inverse fast Fourier transform (IFFT) method to identify the regions giving rise to individual superlattice reflections. Therefore, the regions with different colors (**Figure 2i**) represent nanosized domains with different orientations. Although it is not an approach to directly prove the presence of polar nanoregions (PNRs), we believe that the observed features are PNRs by considering the following aspects:

- The frequency dispersion of the NN9SS_1.0Mn sample is well fitted by the Vogel-Fulcher relation (**Figure 2c**). The PDF analysis shows that the average structure of the NN9SS_1.0Mn sample is pseudocubic, with a small lattice distortion of 1.0014. In contrast, the lattice distortion in the 1-20 Å range is significantly larger (1.0108), indicating the polar nature of the material at nanoscale. These results provide strong evidence that the NN9SS_1.0Mn sample is a relaxor and, therefore, the presence of PNRs is to be expected.
- The local structural heterogeneities, characterized by alternating bright and dark contrasts of a few nanometers in size (**Figure 2f**), are in good agreement with the observed PNRs morphology in the $\text{Na}_{1/2}\text{Bi}_{1/2}\text{TiO}_3$ - BaTiO_3 system (Wohninsland *et al.* Appl. Phys. Lett., **2021**, 118, 072903), which is a well-recognized relaxor material.

c) Nevertheless, we understand that one common practice to confirm that such nanosized regions are PNRs in the literature is to calculate the local polarization using high-resolution scanning transmission electron microscope (STEM) techniques, such as those used for $\text{PbMg}_{1/3}\text{Nb}_{2/3}\text{O}_3$ (Eremenko *et al.* Nat. Commun., **2019**, 10, 2728) and $\text{Na}_{1/2}\text{Bi}_{1/2}\text{TiO}_3$ - BaTiO_3 systems (Fetzer *et al.* Phys. Rev. Mater., **2022**, 6, 064409). It should be noted that in these systems one type of ion is largely displaced and is responsible for the polarization. Therefore, by subtracting the atomic position and calculating the relative displacement of the contributing ions, the polarization of the system can be reliably derived. However, calculating the polarization in NaNbO_3 is less straightforward because all the ions are comparably displaced (Na: 0.18 Å, Nb: 0.13 Å, and O: 0.2 Å, Sakowski-Cowley, Acta Cryst., **1969**, 25, 851-865). The relative displacement of one of these three ions cannot reflect the actual polarization of the system and, therefore, it is not possible to determine the presence of PNRs from the relative displacement mapping in NaNbO_3 .

The above discussions are incorporated into the revised manuscript on page 7:

“Note that a direct characterization of local polarization in NaNbO_3 -based materials is non-trivial, since both cations (Na^+ and Nb^{5+}) and anions (O^{2-}) are largely displaced³⁸, which is different from PbZrO_3 -based materials where Pb^{2+} displacements dominate. Nevertheless, the polar nature of the nanoscale domains on the 1–20 Å scale will be evidenced by the complementary analysis of X-ray pair distribution functions (PDFs) below.”

Furthermore, the reviewer is asking about the nature of the *R* phase. The crystallographic structures of the *P* phase (*Pbcm*, SG No. 57) and the *R* phase (*Pbnm*, SG No. 62) are shown in Fig. R6. The *P* phase is characterized by $\frac{1}{4}$ superlattice reflections due to the quadrupling of the aristotype perovskite structure along the $[001]_{\text{PC}}$ direction. In contrast, the unit cell of the *R* phase corresponds to a $\sqrt{2} \times \sqrt{2} \times 6$ superlattice of the aristotype perovskite structure, giving rise to $\frac{1}{6}$ superlattice reflections. The above explanations are included in the revised Supplementary Information (Supplementary Fig. 13).

Figure R6 (Supplementary Fig. 13). Crystallographic structures of (a) the *P* phase (*Pbcm* space group) and (b) the *R* phase (*Pbnm* space group), viewed along the $[1\bar{1}0]_{PC}$ direction and $[010]_{PC}$ direction, respectively. The unit cells of the *P* phase and *R* phase are highlighted by the dashed lines and the tilting systems between the adjacent octahedral layers are marked.

The last question, whether the *R* phase is an AFE phase, is a very good question (also see our answer to Comment #4 from Reviewer #1). The *R* phase (*Pbnm* space group) in pure NaNbO_3 is regarded as an AFE phase based on the antiparallel atomic displacements of Na and Nb cations (Peel *et al.*, *Inorg. Chem.*, **2012**, 51, 6876). However, we think that NN9SS_1.0Mn is not AFE for the following reasons:

- Despite the presence of antiferrodistortion, no double loops characteristic of a field-induced phase transition has been reported in the *Pbnm* structure. Since the field-induced phase transition cannot be triggered, the *Pbnm* structure cannot be considered an AFE. Therefore, we decided to refer to this phase as the *R'* phase (also see our answer to Comment #4 from Reviewer #1).
- In our opinion, the literature describes the *R* phase as being AFE phase mainly based on the observation of $1/6$ superlattice reflections in the SAED pattern. However, despite the observation of $1/6$ superlattice reflections (Supplementary Fig. 10), we believe that the NN9SS_1.0Mn sample is in the relaxor state rather than the AFE state. The $1/6$ superlattice reflections can be observed in the SAED patterns only at a very local scale and become less visible when taken from a large region (**Figure 3b & 3c**). This is due to the random distribution of nanodomains in different orientations (relaxor behavior), which leads to the streaking feature in the electron diffraction patterns. That is, the NN9SS_1.0Mn sample (also NN7SS_1.0Mn) does not have long-range AFE order that would otherwise lead to the well-defined $1/6$ superlattice reflections. The relaxor nature of the NN9SS_1.0Mn sample is confirmed by a combination of macroscopic electrical measurements (*P-E* loops in

Figure 1e and permittivity in **Figure 2c**), as well as structural microscopic measurements (HRTEM in **Figure 2f, i** & **Figure 3c** and PDF in **Figure 4** & **Table 1**).

(4) From the P-E loop (**Figure 1c**), NN5SS_1.0Mn shows a very small remnant polarization, so how should the large remnant in **Figure 5e,f** be understood? Besides, why the E-removed c_{pc} (**Figure 5e**) decreases compared with the pristine state?

Reply: Thank you for the observations and interesting questions. **Figure 1c** shows the P-E loops of previously E-field loaded samples (2nd cycle, not virgin loops) and, more importantly, only shows the remanent polarization. On the other hand, the measurements in **Figures 5e & 5f** were conducted on virgin samples and depict the changes of the lattice, which are more evidently depicted in sample's macroscopic strain, instead of polarization. **Figures 5e & 5f** indicate that the structure of the material does not completely return to the virgin state after the field-induced antiferroelectric–ferroelectric phase transition, which is consistent with the remanence of the macroscopic longitudinal strain, as shown in **Figure R7**. A volumetric strain of 0.12% (from diffraction) agrees well with a remanent strain of 0.19% along the longitudinal direction (measured macroscopically) in the remanent state, assuming that the transverse strain is negative (Zhang *et al.* Appl. Phys. Lett., **2021**, 118, 132903). Similar behavior is also observed for PbZrO₃-based antiferroelectrics, as shown **Figure R8**. A low remanent polarization is observed, while the remanent strain is larger (0.045 %). We believe that this phenomenon can be explained by the fact that randomly oriented antiferroelectric domains become preferentially oriented after exposure to an electric field, resulting in a remanent strain, as previously suggested by Park *et al.* (J. Appl. Phys., **1997**, 82, 1798). In contrast, the polarization in these domains can still return to the antiparallel configuration and thus does not contribute to the macroscopic polarization. Similar results have recently been reported by Liu *et al.* (Acta Mater., **2020**, 184, 41-49) using *in situ* synchrotron XRD. We added a sentence in the caption of **Figure 1** in the revised manuscript (page 6) to clarify at what conditions the loops were measured.

Figure R7 (Supplementary Fig. 16). Strain hysteresis loop of the NN5SS_1.0Mn sample, recorded along the longitudinal direction in the first electrical cycle.

Figure R8 (Supplementary Fig. 17). **a** Polarization and **b** strain hysteresis loop of a $(\text{Pb}_{0.97}\text{La}_{0.02})(\text{Zr}_{0.75}\text{Sn}_{0.13}\text{Ti}_{0.12})\text{O}_3$ ceramic sample in the 1st and 2nd electrical cycle.

With respect to the question about the reduced lattice parameter, c_{pc} , after the field-induced phase transition, we note that similar behavior was also observed in pure NaNbO_3 (Zhang *et al.*, Appl. Phys. Lett., **2021**, 118, 132903), as shown in Figure R9. We believe that this is a common feature for NaNbO_3 -based antiferroelectrics with the $Pbcm$ space group. It stems from the fact that there is no spontaneous polarization along the c_{pc} direction, leading to anisotropic cation displacements and thus lattice strains in different crystallographic directions when the phase transition is induced by the electric field. The antiparallel atomic displacements of both Na and Nb cations along the $[110]_{pc}$ direction account for the nonpolar nature of the AFE phase. The atomic displacements change from antiparallel in the AFE state to parallel in the field-induced FE state, which is accompanied by an increase in a_{pc} (b_{pc}) due to an expected in-plane expansion (Figure 5e). Since there is no spontaneous polarization along the out-of-plane direction, shrinkage rather than expansion is expected along this direction, resulting in the reduced c_{pc} . As

can be seen in **Figure 5e**, c_{PC} decreases immediately when the phase transition is triggered.

Figure R9. (a) Pseudocubic lattice parameters and (b) primitive cell volume as a function of electric field amplitude, obtained from LeBail fitting using single-phase models. (c) Simultaneously recorded longitudinal (S_{33}) and transverse (S_{11}) strains, used to calculate the volume strain (S_V). The instantaneous strain ratio $-S_{11}/S_{33}$ is shown in the inset. The original figure is found in Zhang *et al.*, Appl. Phys. Lett., **2021**, 118, 132903.

Some minor issues:

(1) The figures are not arranged according to the order they appear in the text, which brings in confusion. Besides, the serial number of SI figures does not match that in the main text (e.g, Figure S8).

Reply: We thank the reviewer for the careful reading. We have carefully checked through the manuscript and corrected the problem.

(2) Some denotations are not defined, e.g, the R phase, and the γ in Figure 5f.

Reply: Thank you for the reminder, the corresponding definitions are now given in the revised manuscript.

Reviewer #3: Reversible field-induced phase transitions define antiferroelectric perovskite oxides and lay the foundation for high energy-storage densities, required for future green technologies. This work demonstrates an approach of adjusting the local structure and defect chemistry to achieve well-defined double P-E loops and a relaxor state with a high energy efficiency of 90%. Various high-tech testing techniques were adopted to revealing the reversible phase transition and structural changes. However, some problems should be addressed. The obtained well-defined P-E loop in NaNbO₃ is interesting and presents the novelty of this work.

Reply: We thank the reviewer for carefully reading the manuscript and providing comments and suggestions, which help us to improve the manuscript.

(1) However, the author related the ill-defined double P-E loop and high remnant polarization to the high leakage current, due to the high concentration of free charges, which stabilize the induced FE state. The author added MnO₂ to act as an electron trap to reduce the amount of charge carriers and improve the resistance, thus produce double hysteresis loop. From this point, how about addition of excess Na element or adding other additives, they can also produce double hysteresis loop? The origin of double hysteresis loop does not clearly addressed yet. So, more solid evidence is required to support the point of view.

Reply: Thanks for the question. We would like to emphasize that excess Na (1 wt.%) was used for the synthesis of the NaNbO₃-SrSnO₃ solid solution to compensate for possible Na evaporation (Popovič *et al.* RSC Adv., **2015**, 5, 76249-76256). However, as can be seen in **Figure 1b**, a large remanence is observed, so Na excess itself is not sufficient. Note that neither the addition of excess Na nor the addition of other additives (e.g., MnO₂) can produce double polarization hysteresis loops in pure NaNbO₃. Fan *et al.* (J. Eur. Ceram. Soc. **2019**, 39, 4712-4718) studied the effect of A-site nonstoichiometry on the phase stability of NaNbO₃ with Na excess and Na deficiency (Na_{1+x}NbO₃, x = -2 to 1 mol%). The stability of the antiferroelectric phase was found to be increased in Na deficient materials, but the transition was irreversible and therefore no double loops were observed. Molak and colleagues studied the impact of Mn addition on the structure and electrical properties of NaNbO₃ crystals and ceramics, and published a series of papers (Molak, Ferroelectrics, **1988**, 80, 27-30, Molak *et al.* Jpn. J. Appl. Phys., **1992**, 31, 3221-3224, Molak *et al.*, J. Phys.: Condens. Matter, **1994**, 6, 6833-6842, and Molak *et al.*, Ferroelectrics, **1995**, 172, 295-298). None of these papers reported any double loops in NaNbO₃. On the contrary, ferroelectric properties were induced by the addition of Mn when its concentration was higher than 0.4 wt.% (Molak, Ferroelectrics, **1988**, 80, 27-30).

We would like to emphasize here that the results presented in our manuscript clarify the origin of the double hysteresis loop from three different aspects:

- a) **Design of new antiferroelectrics with reversible transition:** A thermodynamic way of thinking about the development of antiferroelectrics is to ensure that the free energies of the polar and antipolar phases are comparable (Rabe, Functional metal oxides: new science

and novel applications, **2013**, 221-244 and Setter, *Ferroelectrics*, **2016**, 500, 164-182). Nevertheless, the search for compositions with a reversible transition is still experimentally difficult, so that only few material systems are known. To address the irreversible transition of pure NaNbO_3 , the basic $\text{NaNbO}_3\text{-SrSnO}_3$ solid solution utilized in this work was designed using first-principles calculations to stabilize the AFE order relative to the FE order (Zhang *et al.*, *Chem. Mater.*, **2021**, 33, 266-274), which differs from traditional approaches based on adjusting the tolerance factor.

- b) **Suppression of remanent polarization by tailoring the defect chemistry:** The stabilization of the AFE order does not necessarily guarantee a well-defined double loop with minimal remanent polarization in NaNbO_3 , as observed in PbZrO_3 -based AFEs, and the reasons remain unknown. Based on the frequency dependence of dielectric loss, we hypothesize that reducing the concentration of mobile charge carriers is the key to suppressing remanent polarization. By adding MnO_2 to the newly designed $\text{NaNbO}_3\text{-SrSnO}_3$ system, we have succeeded for the first time in presenting a well-defined double loop with a very low remanence in NaNbO_3 , whose remanent polarization is 14 times smaller than that of pure NaNbO_3 .
- c) **Revealing the structural origin underlying the double loop:** We used *in situ* electric field synchrotron XRD to reveal the presence of AFE state, field-induced FE state, and remanent state for the NN5SS_1.0Mn sample studied, indicating the reversible nature of the transition. Both Na and Nb cations contribute to the local polarizations of the *Pbcm* structure (Figure R10). When the field-induced phase transition is triggered, the atomic displacements of Na and Nb cations change from antiparallel to parallel, resulting in an increase in a_{PC} , b_{PC} , and γ values (Figure 5e & 5f). Our detailed structural analysis shows that the atomic displacements of the Na(1) ions relative to the symmetric centers in the cubic phase are 0.103 Å and 0.025 Å in NaNbO_3 and NN5SS_1.0Mn samples (Supplementary Table. 5), respectively. As a result, the orthorhombic lattice distortion decreases from 1.0113 for NaNbO_3 to 1.0085 for NN5SS_1.0Mn. This has two consequences: First, the critical field required to trigger the transition increases from 12 kV/mm for NaNbO_3 to 23 kV/mm for NN5SS_1.0Mn, as it becomes more difficult to break the centrosymmetric structure with a smaller lattice distortion; second, the rigid structure is more likely to return the field-induced structure to its original state, providing the restoring force for the reversible phase transition.

The discussions regarding the origin of reversible structural transition are now included on page 13 in the revised manuscript:

“The cell parameters a_{PC} , b_{PC} , γ , and cell volume show a significant increase at the origin of the phase transition due to the change in the atomic displacements of the Na and Nb cations from antiparallel to parallel. Structural analysis reveals that the atomic displacements of the Na(1) ions relative to the symmetric centers in the cubic phase are 0.103 Å and 0.025 Å in the NaNbO_3 and NN5SS_1.0Mn samples (Supplementary Table. 5), respectively. As a result, the orthorhombic lattice distortion decreases from 1.0113 for NaNbO_3 to 1.0085 for NN5SS_1.0Mn. This has two consequences: First, the critical field required to trigger the transition increases, as it becomes more difficult to break the centrosymmetric structure; second, the rigid structure is more likely to return the field-induced structure to its original state and provide the restoring force for the reversible phase transition.”

Figure R10 (Supplementary Fig. 19). A schematic of the crystallographic structure of the NaNbO_3 -based materials with the $Pbcm$ space group along the $[001]_{PC}$ direction. The orientation of the double-headed arrow represents the direction of the antiparallel atomic displacement of Nb atoms.

(2) The author paid great efforts to produce a reversible AFE-FE phase transition and stated the AFE phase possess a $Pbcm$ space group (NN5SS_1.0Mn), which should be a P phase. However, it is found the slim P-E loops with increased energy storage density and efficiency are observed in NN9SS_1.0Mn, which possesses an R phase with a $Pbnm$ space group. These two compositions are of totally different phase structures. So, the question come out that what's the effect of the well-defined P-E loops in NN5SS_1.0Mn for the slim P-E loops in NN9SS_1.0Mn and the improved energy storage density and efficiency. It's more interesting to get a relaxor character in the P phase. From the viewpoint of energy storage performance, ultrahigh energy storage density with value over 10 J/cm^3 was already reported in R phase in many NaNbO_3 -based materials in the literatures, more than 5 times higher than the value of this work, and the mechanism of relaxor feature was also well-explained (Adv. Funct. Mater.2019, 29, 1903877). So, the concept is not new and the obtained energy storage properties are poor.

Reply: Thank you for this remark. First, we wish to answer the question regarding the comparison of NN5SS_1.0Mn and NN9SS_1.0Mn. These two compositions are important, as they represent the two different behaviors that we can achieve in this material – AFE phase with field-induced transition and abrupt property change (e.g., polarization or strain) on one hand, and relaxor composition without phase transition and slim hysteresis on the other hand. Note that in this manuscript we focus on providing basic scientific understanding for such behavior and study the mechanisms and structures, which represent the main novelty of this work. We wish to demonstrate a general concept for material design from the crystal-chemical and defect chemistry perspective. We provide new understandings towards phase transitions, (micro)structural changes, and defect chemistry, which shed light on the underlying mechanisms of the interesting antiferroelectricity phenomenon. Nevertheless, we also strongly believe that these materials have a large potential for applications – this of course depends on the application type. The AFE-based compositions (e.g., NN5SS_1.0Mn) have a good energy storage density, but their main advantage is the fast charge release due to the transition. This is of interest for pulse applications or capacitors operating at intermediate/low frequencies. On the other hand, the relaxor compositions (e.g., NN9SS_1.0Mn) have good energy storage characteristics and high efficiency, making them useful for capacitors operating at higher frequencies.

With respect to the magnitude of the energy storage density, we agree with the reviewer that some earlier publications have reported higher absolute values. However, we would like to note that extremely high values have mostly been achieved because the samples were loaded with a higher E -field. Energy storage density namely scales with increasing applied electric field and should therefore be reported together with the field magnitude (or in a normalized/relative manner). The applied electric field for the materials in our work (~ 20 kV/mm) is 2-5 times smaller than that reported in the literature. In fact, we have discussed this issue in one of our recent publications (Zhang *et al.*, *J. Materiomics*, **2022**, <https://doi.org/10.1016/j.jmat.2022.09.008>) and were comparing some of the prominent literature reports in a normalized manner (Figure R11). As can be seen, all reports lie more or less on the same line, indicating that the energy storage density mostly depends on the achieved breakdown field. The latter was in our case between 18-25 kV/mm, which is already very high for polycrystalline bulk ceramics. However, this value can easily be increased by producing the material in multilayered capacitor form (with layer thickness around 20-80 μm) or even as thin film. The previously-reported high energy densities are thus mostly related to the material form/geometry and not to the material itself. We also note that in our case we used symmetric

electrodes that covered the whole sample area, which has not been done in many of the previous reports with high energy density values (the effective electric fields applied to those samples can therefore strongly deviate from the applied fields!).

Figure R11. Electric field dependence of the energy storage density (Zhang *et al.* J. Materiomics., 2022, <https://doi.org/10.1016/j.jmat.2022.09.008>). The data were obtained from refs. [1-10].

The reviewer also mentioned that it may be interesting to induce relaxor behavior in the P phase. However, we have never observed this (in our work or literature) and we question if one can get relaxor behavior in the P phase, since the defining characteristics of relaxors, i.e., compositional and/or structural inhomogeneities at the nanoscale, seem to contradict the defining structural features of the $Pbcm$ space group (ordered superlattice structures with a length scale of 1.6 nm in the unit cell, antiparallel displacements, complex oxygen octahedral tilting). We thus believe that the inducement of the relaxor state in NaNbO_3 is inevitably related to a change of the P phase into another symmetry.

(3) What's more, this manuscript even cannot induce an AFE-FE phase transition in NN9SS_1.0Mn, as said “no field-induced phase transition is observed for the NN9SS_1.0Mn composition under the present experimental conditions”, in this case, high energy efficiency with low energy storage density is always observed. We even cannot evaluate the change of P-E loops and other electrical properties once the AFE-FE phase transition is induced, such as the hysteresis, remnant polarization, maximum polarization and the energy storage performance. So, I feel this manuscript cannot meet the standard of Nature Communications.

Reply: The reviewer is correct, there is no field-induced transition in the NN9SS_1.0Mn sample, since this sample does not exhibit the AFE phase (more details are given in our answer to the Comment #4 from Reviewer #1). In fact, this is one of the important findings of our work. It is not our intention to state that high energy densities require a field-induced transition. With our

work we demonstrate that:

- I) minor chemical modifications of NaNbO_3 can induce different phases and thus different behavior (AFE or relaxor);
- II) comparable energy density values can be observed in both cases, in AFE compositions (1.70 J cm^{-3} for NN5SS_1.0Mn) and relaxor compositions (1.75 J cm^{-3} for NN9SS_1.0Mn).

The mechanism for energy storage is different, as seen from the different shape of the P - E loops (**Figure 1**). We note that this finding is nontrivial, since many recent publications on lead-free energy storage materials claim that their compositions are antiferroelectric and thus high energy storage values should be related to the field-induced transition, but here we demonstrate that this is not a precondition and therefore detailed structural studies are always required. We hope to shed light on these open questions with our work.

We believe that the second part of this comment from the reviewer (regarding not being able to evaluate the change of P - E loops and other electrical properties) is related to some misunderstanding. We have conducted extensive electrical measurements on all the reported compositions, including dielectric measurements (**Figure 1 & Figure 2**) and high-voltage polarization measurements (**Figure 1** and Supplementary Figs. 4, 5 & 6). These measurements can be evaluated using conventional procedures, established in the ferroelectrics community. The evaluated parameters are provided in Table R2 below.

Table R2 (Supplementary Table 7). Electrical Properties of NN5SS, NN5SS_1.0Mn, and NN9SS_1.0Mn samples.

	NN5SS	NN5SS_1.0Mn	NN9SS_1.0Mn
ϵ (RT, @ 10 kHz)	345	325	1444
P_r ($\mu\text{C cm}^{-2}$)	11.0	3.2	0.8
P_m ($\mu\text{C cm}^{-2}$)	31.1	30.1	20.0
	@ 16 kV mm^{-1}	@ 23 kV mm^{-1}	@ 25 kV mm^{-1}
W_{stor} (J cm^{-3})	4.2	5.2	2.0
W_{rec} (J cm^{-3})	0.90	1.70	1.75
η (%)	21	33	90

ϵ : dielectric permittivity, P_r : remanent polarization, P_m : polarization at the maximum electric field, W_{stor} : storage energy density, W_{rec} : recoverable energy density and η : energy-storage efficiency ($W_{\text{stor}}/W_{\text{rec}}$).

References

- [1] H. Qi, R. Zuo, A. Xie, A. Tian, J. Fu, Y. Zhang, S. Zhang, Ultrahigh Energy-Storage Density in NaNbO_3 -Based Lead-Free Relaxor Antiferroelectric Ceramics with Nanoscale Domains, *Adv. Funct. Mater.* 29(35) (2019) 1903877.
- [2] M. Zhou, R. Liang, Z. Zhou, S. Yan, X. Dong, Novel Sodium Niobate-Based Lead-Free Ceramics as New Environment-Friendly Energy Storage Materials with High Energy Density, High Power Density, and Excellent Stability, *ACS Sustainable Chemistry & Engineering* 6(10) (2018) 12755-12765.
- [3] X. Dong, X. Li, X. Chen, H. Chen, C. Sun, J. Shi, F. Pang, H. Zhou, High energy storage density and power density achieved simultaneously in NaNbO_3 -based lead-free ceramics via antiferroelectricity enhancement, *Journal of Materiomics* 7(3) (2021) 629-639.
- [4] A. Xie, R. Zuo, Z. Qiao, Z. Fu, T. Hu, L. Fei, NaNbO_3 - $(\text{Bi}_{0.5}\text{Li}_{0.5})\text{TiO}_3$ Lead-Free Relaxor Ferroelectric Capacitors with Superior Energy-Storage Performances via Multiple Synergistic Design, *Advanced Energy Materials* 11(28) (2021).
- [5] A. Xie, J. Fu, R. Zuo, Achieving stable relaxor antiferroelectric P phase in NaNbO_3 -based lead-free ceramics for energy-storage applications, *Journal of Materiomics* 8(3) (2022) 618-626.
- [6] Z. Chen, S. Mao, L. Ma, G. Luo, Q. Feng, Z. Cen, F. Toyohisa, X. Peng, L. Liu, H. Zhou, C. Hu, N. Luo, Phase engineering in NaNbO_3 antiferroelectrics for high energy storage density, *Journal of Materiomics* 8(4) (2022) 753-762.
- [7] J. Jiang, X. Li, L. Li, S. Guo, J. Zhang, J. Wang, H. Zhu, Y. Wang, S.-T. Zhang, Novel lead-free NaNbO_3 -based relaxor antiferroelectric ceramics with ultrahigh energy storage density and high efficiency, *Journal of Materiomics* 8(2) (2022) 295-301.
- [8] J. Liu, P. Li, C. Li, W. Bai, S. Wu, P. Zheng, J. Zhang, J. Zhai, Synergy of a Stabilized Antiferroelectric Phase and Domain Engineering Boosting the Energy Storage Performance of NaNbO_3 -Based Relaxor Antiferroelectric Ceramics, *ACS Appl. Mater. Interfaces* 14(15) (2022) 17662-17673.
- [9] A. Xie, J. Fu, R. Zuo, C. Zhou, Z. Qiao, T. Li, S. Zhang, NaNbO_3 - CaTiO_3 lead-free relaxor antiferroelectric ceramics featuring giant energy density, high energy efficiency and power density, *Chem. Eng. J.* 429 (2022) 132534.
- [10] H. Qi, R. Zuo, A. Xie, J. Fu, D. Zhang, Excellent energy-storage properties of NaNbO_3 -based lead-free antiferroelectric orthorhombic P-phase (Pbma) ceramics with repeatable double polarization-field loops, *J. Eur. Ceram. Soc.* 39(13) (2019) 3703-3709.

REVIEWERS' COMMENTS

Reviewer #1 (Remarks to the Author):

(1) What are the noteworthy results?

 Yes, after the comments are addressed, now the results are noteworthy.

(2) Will the work be of significance to the field and related fields? How does it compare to the established literature? If the work is not original, please provide relevant references.

 Yes, definitely this work would have a significant impact in the field of energy storage, as NaNbO_3 is the potential candidate for next-generation energy storage applications. The authors in this work have produced significant results for searching for new energy storage material which in this work is NaNbO_3 AFE that exhibits double polarization hysteresis loops at ambient conditions.

(3) Does the work support the conclusions and claims, or is additional evidence needed?

 After revision now the work supports the conclusion and claims and no additional evidence is needed.

(4) Are there any flaws in the data analysis, interpretation, and conclusions? Do these prohibit the publication or require revision?

 There are no flaws in the data analysis, interpretation, and conclusions. Now the manuscript (after revision) can be published.

(5) Is the methodology sound? Does the work meet the expected standards in your field?

 Definitely, as NaNbO_3 -based solid solutions are a real need in energy storage applications.

(6) Is there enough detail provided in the methods for the work to be reproduced?

 Yes

Reviewer #2 (Remarks to the Author):

The authors have addressed all my concerns/questions. I am happy to recommend its publication in Nature Communications.

Reviewer #3 (Remarks to the Author):

All questions are addressed, it is suggested to be accepted.